# Genetic associations between non-cognitive skills and academic achievement over development

**Margherita Malanchini** [1,2,14] ✉, **Andrea G. Allegrini** [2,3,14] ✉,
**Michel G. Nivard** [4], **Pietro Biroli** [5], **Kaili Rimfeld** [2,6], **Rosa Cheesman** [7],
**Sophie von Stumm** [8], **Perline A. Demange** [4,7,9,10], **Elsje van Bergen** [4,9,10],
**Andrew D. Grotzinger** [11], **Laurel Raffington** [12], **Javier De la Fuente** [13],
**Jean-Baptiste Pingault** [2,3], **Elliot M. Tucker-Drob** [13], **K. Paige Harden** [13] &
**Robert Plomin** [2]

Non-cognitive skills, such as motivation and self-regulation, are partly heritable and predict academic achievement beyond cognitive skills. However, how the relationship between non-cognitive skills and academic achievement changes over development is unclear. The current study examined how cognitive and non-cognitive skills are associated with academic achievement from ages 7 to 16 years in a sample of over 10,000 children from England and Wales. The results showed that the association between non-cognitive skills and academic achievement increased across development. Twin and polygenic scores analyses found that the links between non-cognitive genetics and academic achievement became stronger over the school years. The results from within-family analyses indicated that non-cognitive genetic effects on academic achievement could not simply be attributed to confounding by environmental differences between nuclear families, consistent with a possible role for evocative/active gene–environment correlations. By studying genetic associations through a developmental lens, we provide further insights into the role of non-cognitive skills in academic development.

Children who are emotionally stable, motivated and capable of regulating their attention and impulses do better in school, independent of their level of cognitive (Cog) ability[1–7]. These important socio-emotional characteristics have been broadly described as non-cognitive skills (NCS)[8]. 'Non-cognitive' is an imperfect term that primarily serves to differentiate these characteristics from what they are not – performance on standardized tests of cognitive ability. The panoply of non-cognitive skills that predict better educational outcomes can be organized into three partly overlapping domains: motivational factors, self-regulatory strategies and personality traits[9].

Twin research has shown that genetic differences between people contribute to their differences in non-cognitive skills.

Most domains of non-cognitive skills, including academic motivation[10,11], self-regulation[12] and personality[13], are moderately heritable (~30–50%). In addition, twin studies have found evidence that non-cognitive skills are genetically correlated with academic achievement[14,15]. That is, some of the same genetic differences that are associated with variation in academic achievement are also associated with non-cognitive skills.

DNA-based methods have confirmed genetic links between non-cognitive skills and academic performance. Genome-wide association studies (GWAS) of educational attainment (EA) (that is, years of formal education completed) have identified genetic variants that are correlated with completing formal education[16,17]. A polygenic score

(PGS) constructed from these GWAS results predicts higher levels of self-control[18], more adaptive personality traits (higher conscientiousness, agreeableness and openness to experience) and greater academic motivation[19]. Additionally, previous GWAS work has identified associations between DNA variants and EA that were independent of cognitive test performance, essentially performing a GWAS of non-cognitive skills[20]. The genetics of non-cognitive skills were found to be related to conscientiousness, openness to experience, delay of gratification and health-risk behaviours[20].

The current study uses both twin and DNA-based methods to expand our understanding of the association between non-cognitive skills and academic achievement over development. We address four key questions (Fig. 1). First, does the strength of the association between non-cognitive skills and academic achievement change over development (from age 7 to age 16 years)? Second, do genetic dispositions towards non-cognitive skills vary in their association with academic achievement across development? Third, to what extent are these associations accounted for by between-family processes, such as environmental influences shared between individuals in a family? Fourth, do non-cognitive genetic associations with academic achievement vary by socio-economic status (SES)?

First, we investigated the links between non-cognitive skills and academic achievement across the school years. Developmental studies that have investigated the association between non-cognitive skills and academic achievement remain scarce and have focused on a few specific measures over relatively short time frames[21]. Here, we analyse a comprehensive battery of developmental data collected from over 10,000 children born in England and Wales who were followed across compulsory education (Fig. 1, left). Furthermore, we simultaneously consider the role of cognitive skills in these associations. Past research has highlighted how skills that are broadly considered non-cognitive, such as self-control, rely on cognitive competencies[22]. Therefore, it is important to take into account developing cognitive skills when assessing the relationship between non-cognitive skills and academic achievement over time.

Second, we investigated whether genetic dispositions towards non-cognitive skills become increasingly predictive of academic achievement across development. Twin studies focusing on specific moments in childhood or adolescence[23] have found that heritable variation in non-cognitive skills, such as motivation and self-regulation, contribute to academic achievement beyond cognitive competencies[24]. However, this relationship across development remains underinvestigated. We triangulate evidence across different methods, including twin and PGS analyses, to investigate the association between genetic factors linked to cognitive and non-cognitive skills and academic achievement across compulsory education.

Third, with a sibling-difference design, we examined to what extent the developmental relationship between genetic propensity for non-cognitive skills and academic achievement is accounted for by family-wide environmental processes. Sibling differences in genotypes are randomized by meiosis, such that siblings have an equal probability of inheriting any given parental allele. Therefore, within-sibling pair PGS associations are thought to be less confounded by environmental differences between nuclear families, including population stratification and indirect genetic effects[25]. Indirect genetic effects refer to the association between the non-transmitted parental genotypes and the offspring phenotype, potentially reflecting rearing environments, although they can also capture broader demographic phenomena, such as assortative mating[26].

Conversely, differences between siblings in PGS associations are often referred to as 'direct' genetic effects[27] in that they are consistent with a causal effect of genetic variants within an individual on their phenotype. However, even direct genetic effects involve mediation through environmental processes. For example, children with a greater motivation towards academic achievement might actively select, modify and

create environmental experiences that foster further achievement, such as deciding to take advanced classes[28]. That is, genetic differences between children can result in differential exposure to learning environments, which, in turn, can affect their academic achievement[29]. These active/evocative gene–environment correlations amplify the effects of genetic difference and are one theorized mechanism for increasing genetic associations over development[30,31].

Fourth, we explored whether genetic associations with academic achievement varied by SES. Genetic and environmental processes might interact such that the effects of environmental experiences on a trait might be partly dependent on genetic factors and vice versa[32,33]. Studies that examined this possibility have focused on the role of socio-economic disadvantage across a broad range of contexts, including family SES[34,35] and the school environment[36,37]. We explore whether the cognitive and non-cognitive PGS prediction of academic achievement differs at different levels of socio-economic disadvantage across development.

Under a developmental lens, these analyses address four core research questions providing a detailed account of the processes through which cognitive and non-cognitive skills are linked to individual differences in academic achievement. We triangulated evidence across multiple genetic methods. Since each method is subject to different and unrelated assumptions and limitations, triangulating multiple methods provides a powerful tool to increase the reliability of our results[38].

## Results

### Associations between non-cognitive skills and academic achievement

Parents, teachers and twins rated different non-cognitive skills at different ages. On the basis of extant literature and measures availability, we focused on two broad dimensions of non-cognitive skills, which were modelled as latent factors (Fig. 1): (1) education-specific non-cognitive skills, including measures of academic interest, attitudes towards learning and academic self-efficacy and (2) domain-general self-regulation skills, including measures of behavioural and emotional regulation not necessarily related to the school context (Fig. 1 and Methods). Here, we report analyses of these two dimensions. Analyses of individual measures are reported in Supplementary Information (Supplementary Note 1 and Supplementary Tables 1b and 2).

Latent factors of education-specific non-cognitive skills and domain-general self-regulation skills (Supplementary Tables 3 and 4) were positively correlated with academic achievement at all developmental stages. Effect sizes differed by rater and developmental stage and tended to increase with age. For example, the association between self-rated education-specific non-cognitive skills and academic achievement increased from $r = 0.10$ (95% confidence interval (CI) 0.07 to 0.14) at 9 years of age, to $r = 0.41$ (95% CI 0.38 to 0.44) at 12 years of age and to $r = 0.51$ (95% CI 0.48 to 0.55) at 16 years of age (Supplementary Note 1, Supplementary Fig. 1 and Supplementary Table 5). Latent non-cognitive factors were also modestly correlated with latent factors of general cognitive ability (Supplementary Table 6) at the same age (Supplementary Table 7).

We examined whether general cognitive ability could account for the associations between non-cognitive skills and academic achievement. Multiple regression analyses showed that both non-cognitive factors were substantially and significantly associated with academic achievement beyond cognitive skills at every stage of compulsory education (Fig. 2a and Supplementary Table 8). The relative association between non-cognitive skills and academic achievement increased developmentally, particularly when considering self-reported measures. For self-reported education-specific non-cognitive skills, the effect size of the relative prediction of achievement increased from $\beta = 0.10$ (95% CI 0.06 to 0.13) at 9 years of age (effect size for Cog ability: $\beta = 0.46$, 95% CI 0.44 to 0.48) to $\beta = 0.28$ (95% CI 0.24–0.32) at

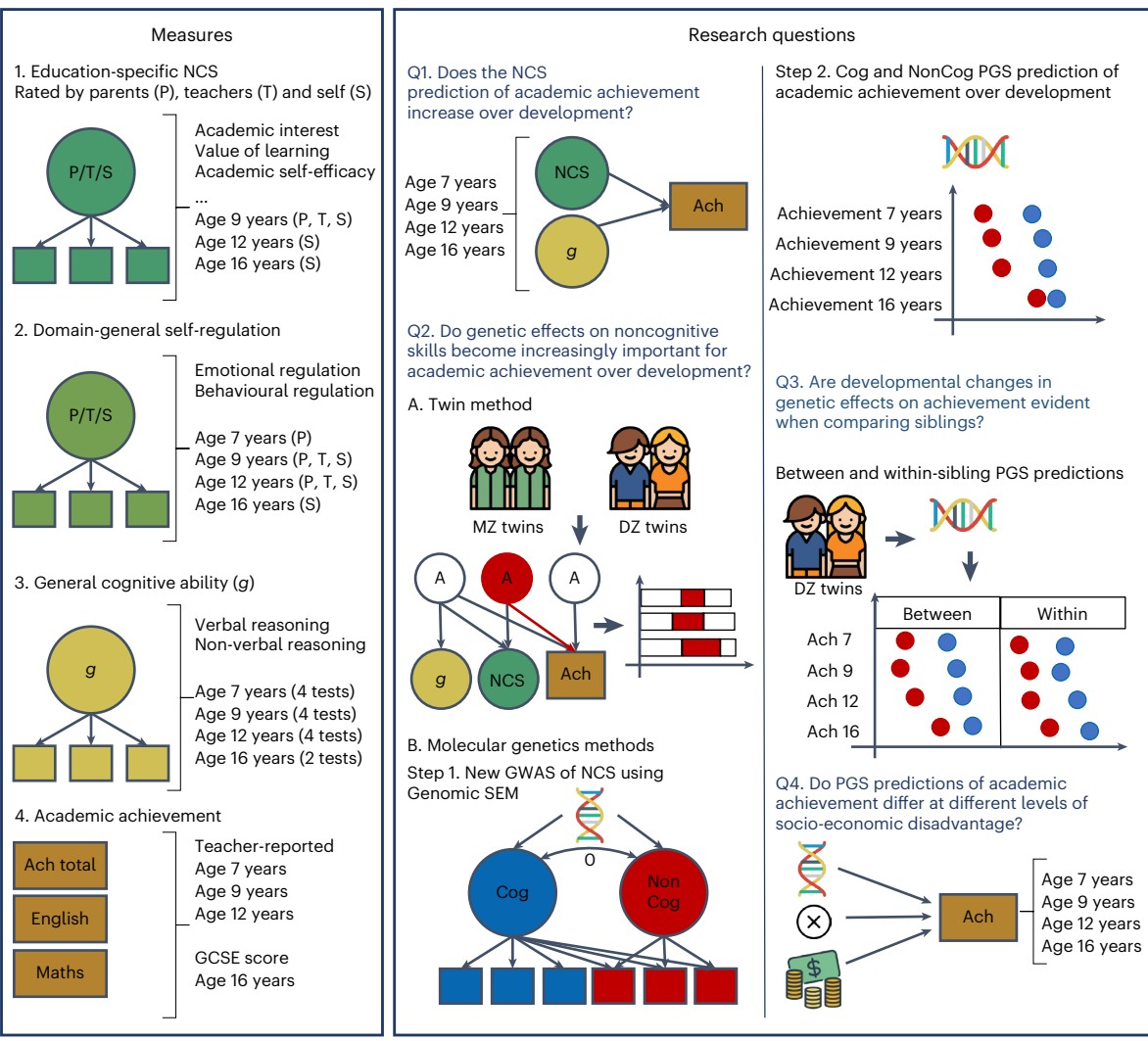

**Fig. 1 | A visual summary of the measures, research questions and methods adopted in the present study.** Left: we used factor analysis to capture individual differences in two broad dimensions of non-cognitive skills (NCS): education-specific NCS (including measures such as academic interest, academic self-efficacy and value attributed to learning) and domain-general self-regulation skills (including measures of behavioural and emotional regulation not necessarily related to the school context). We also created latent measures of general cognitive ability (g) from verbal and non-verbal tests at four ages. Academic achievement (Ach) measures included teacher ratings of academic performance based on the national curriculum at ages 7, 9 and 12 years and exam scores at 16 years (see Methods for a detailed description). Centre and right: a summary of the methodologies adopted to address each of the four core research questions in the study. We addressed the first research question (Q1) by conducting a series of multiple regressions to investigate changes in the developmental contribution of NCS to academic achievement beyond cognitive skills. We addressed the second research question (Q2) using multiple genetic methods. First (A), we conducted trivariate Cholesky decompositions using twin data. Second (B), we created a GWAS of NCS by extending the GWAS-by-subtraction[20] approach with a set of GWAS for specific cognitive tasks and SES-relevant traits and examined developmental changes in the cognitive (Cog) and non-cognitive (NonCog) polygenic score (PGS) prediction of academic achievement from age 7 to 16 years. We addressed our third research question (Q3) by modelling Cog (blue) and NonCog (red) PGS effects within a sibling difference design, therefore separating within-family from between-family effects. We investigated our fourth research question (Q4) fitting multivariable models, including the effects of the Cog/NonCog PGS, family SES and their two-way interaction.

12 years of age (effect size for Cog ability: $\beta = 0.36$, 95% CI 0.32 to 0.40) to $\beta = 0.58$ (95% CI 0.54 to 0.62) at 16 years of age (effect size for Cog ability: $\beta = 0.39$, 95% CI 0.37 to 0.41). A developmental increase was also observed for self-reported measures of domain-general self-regulation skills, for which the predictive power increased from $\beta = 0.11$ (95% CI 0.07 to 0.15) at 9 years of age to $\beta = 0.21$ (95% CI 0.19 to 0.23) at 16 years of age, after accounting for general cognitive ability (Supplementary Table 8).

**Univariate and multivariate twin analyses**

Applying twin designs (Methods), we found that the heritability (that is, the extent to which observed differences in a trait are accounted for by genetic differences) of non-cognitive skills differed significantly across raters and developmental stages (Supplementary Note 2, Supplementary Table 9 and Supplementary Figs. 2–6). Heritability estimates of latent non-cognitive factors, which exclude error of measurement, ranged between 70% (95% CI 0.63 to 0.77) for self-reported education-specific skills at 9 years of age and 93% (95% CI 0.91 to 0.96) for parent-reported education-specific non-cognitive skills at 9 years of age (Supplementary Note 2, Supplementary Tables 10 and 11 and Supplementary Fig. 7). These substantial heritability estimates are consistent with previous studies that investigated the heritability of latent dimensions of non-cognitive skills[11] and of a general factor of psychopathology across different raters[39]. The correlation between non-cognitive measures and academic achievement was mostly accounted for by genetic factors and, to a lesser extent, by

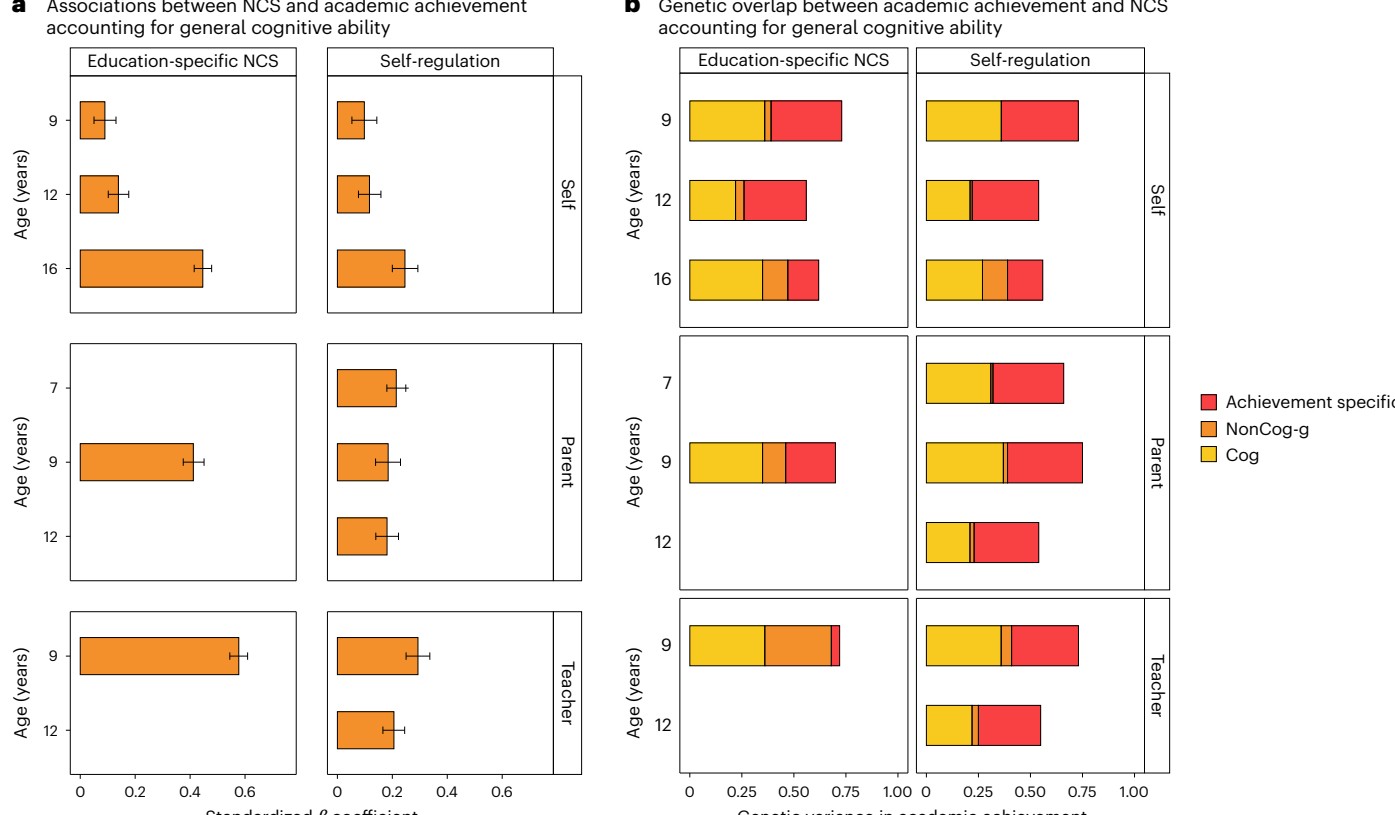

**Fig. 2 | Associations between non-cognitive skills and academic achievement accounting for general cognitive ability. a**, Associations between latent factors of non-cognitive skills (NCS) and academic achievement at ages 7, 9, 12 and 16 years, after accounting for general cognitive ability at the same age using multiple regression. Each bar indicates the effect size of standardized regression coefficients and the error bars indicate the 95% CI around the estimates. The left side shows the associations for latent measures of education-specific NCS, while the right shows the associations for latent dimensions of domain-general self-regulation skills. The figure is further divided into self-rated (top), parent-rated (middle) and teacher-rated (bottom) measures. *N* of independent samples ranged between 1,742 and 3,843; the exact *N* for each regression analysis are reported in Supplementary Table 8. **b**, Each bar represents genetic

effects (standardized and squared path estimates) on academic achievement over development and includes three shadings. The lighter (yellow) shading indicates the proportion of genetic variance in academic achievement that can be attributed to genetic variance in general cognitive ability (g). The orange shadings indicate the proportion of genetic variance in academic achievement that can be attributed to genetic variance in NCS, independent of the genetics of cognitive skills (NonCog-g). The red shading indicates genetic effects on academic achievement independent of the genetics of cognitive and non-cognitive skills (achievement specific). The results are further divided into self-rated (top), parent-rated (middle) and teacher-rated (bottom) measures. Standardized paths and 95% CI for all estimates are presented in Supplementary Tables 12 and 13.

non-shared environmental factors (Supplementary Note 2 and Supplementary Fig. 6).

We then investigated whether the observed genetic associations between latent non-cognitive factors and academic achievement could be accounted for by genetic factors associated with cognitive skills. We investigated this question with a series of trivariate Cholesky decompositions (Methods), the results of which are presented in Fig. 2b, which reports standardized squared path estimates, and Supplementary Tables 12 and 13, which report standardized path estimates and 95% CIs. Similar to hierarchical regression, the Cholesky approach parses the genetic and environmental variation in each trait into that accounted for by traits previously entered into the model and the variance which is unique to a newly entered trait.

Each bar in Fig. 2b is the outcome of a different trivariate Cholesky decomposition of the heritability of academic achievement (the total length of the bar) into genetic effects associated with non-cognitive skills after controlling for genetic effects associated with cognitive skills at the same age. We found that genetic effects associated with cognitive skills accounted for between 21% and 36% of the total variance in academic achievement, as indicated by standardized paths ranging between 0.46 (95% CI 0.37 to 0.54) and 0.60 (95% CI 0.50 to 0.70). Genetic effects associated with non-cognitive skills, independent of

cognitive skills, accounted for between 0.1% and 32.5% of the variance in academic achievement. Standardized paths ranged between 0.01 (95% CI −0.16 to 0.17) for self-reported self-regulation at 9 years of age and 0.57 (95% CI 0.48 to 0.67) for teacher reported education-specific non-cognitive skills at 9 years of age. Last, we found that between 5% and 37% of the variance in academic achievement was independent of genetic effects associated with cognitive and non-cognitive skills. Standardized paths ranged between 0.23 (95% CI 0.13 to 0.33) and 0.61 (95% CI 0.52 to 0.70).

The top three rows of Fig. 2b illustrate the developmental increase in how the genetics of self-reported non-cognitive skills contribute to the genetics of academic achievement. Focusing on education-specific non-cognitive skills, we found that standardized squared path estimates increased from explaining 1% of the total variance in academic achievement at 9 years of age (standardized path estimate of 0.01, 95% CI −0.16 to 0.17) to 4% at 12 years of age (standardized path estimate of 0.16, 95% CI 0.02 to 0.30) and 12% of the total variance in achievement at 16 years of age (standardized path estimate of 0.35, 95% CI 0.26 to 0.44) (Supplementary Tables 12 and 13). This increased contribution beyond cognitive skills was also observed for domain-general self-regulation. See Supplementary Fig. 7 for the full models' results, which include shared and non-shared environmental estimates.

## A PGS of non-cognitive skills

To obtain a PGS for use in subsequent analyses, we first extended previous work using the GWAS-by-subtraction approach to identify genetic variants associated with non-cognitive skills[20]. Previous GWAS-by-subtraction work leveraged genomic structural equation modelling (SEM)[40] and the two GWAS of EA and cognitive performance (CP) to separate the genetic variance in EA into a cognitive component and a residual non-cognitive component. We extended this model in two directions. First, we extended the latent cognitive factor by including GWAS summary statistics from additional cognitive measures (episodic memory; processing speed, executive functions and reaction time)[41]. Second, we included other socio-economic attainment variables, including Townsend Deprivation and Income[42], in addition to EA[17]. The resulting non-cognitive factor can therefore be defined as genetic variation shared by EA, income and neighbourhood deprivation that is independent of all measured cognitive abilities. Akin to Demange et al.[20], we then fitted a Cholesky model (Methods) where indicators of the non-cognitive latent factor (henceforth NonCog) were regressed on the cognitive latent factor (henceforth Cog; Fig. 3a and Supplementary Table 14).

Our Cog and NonCog factors correlated strongly with those obtained from Demange et al.[20] (Supplementary Table 15). The genetic correlation ($r_g$) was 0.96 between the Cog factors and 0.93 between the NonCog factors. The $r_g$ between Cog and NonCog was 0.15. Supplementary Fig. 8 shows the $r_g$ values between the Cog and NonCog genetics and 18 psychiatric, personality and socio-economic traits, which we compared with the $r_g$ values obtained by Demange et al.[20]. The pattern of associations was largely consistent across the two models. However, in some instances, the results diverged. Specifically, with respect to psychiatric traits, autism, anorexia and attention deficit/hyperactive disorder, a larger gap was observed between the Cog and NonCog factors compared to Demange et al., where differences in the correlations were less pronounced or absent. As expected, the results differed most for socio-economic traits, with stronger correlations for NonCog than Cog with longevity ($r_g = 0.52$, 95% CI 0.44 to 0.60, $P = 1.04 \times 10^{-45}$ versus $r_g = 0.35$, 95% CI 0.29 to 0.41, $P = 6.40 \times 10^{-31}$), neighbourhood DE ($r_g = -0.66$, 95% CI −0.74 to −0.58, $P = 3.85 \times 10^{-54}$ versus $r_g = -0.28$, 95% CI −0.36 to −0.21, $P = 5.98 \times 10^{-12}$) and EA ($r_g = 0.83$, 95% CI 0.81 to 0.85, $P = 0.00$ versus $r_g = 0.65$, 95% CI 0.63 to 0.67, $P = 0.00$; Supplementary Fig. 8 and Supplementary Table 15).

## NonCog PGS effects increase developmentally

We calculated PGS for Cog and NonCog and examined their association with cognitive, non-cognitive and academic phenotypes over development. PGS analyses leverage findings from GWAS and aggregate single-nucleotide polymorphisms (SNPs) across the genome into a single composite index that summarizes genetic influence on a target trait. We calculated PGS as the sum of SNPs at all loci weighted by the effect size of their association (Methods). We first investigated whether and to what extent Cog and NonCog PGS predicted individual differences in non-cognitive skills across development by modelling both PGSs in a multiple regression model (Methods).

In line with our previously obtained results showing a moderate association between Cog and NonCog traits, we found that the Cog PGS significantly predicted variation in non-cognitive skills across development, with standardized effect sizes ranging between $\beta = 0.04$ (95% CI 0.001 to 0.079) and $\beta = 0.22$ (95% CI 0.18 to 0.26; Supplementary Fig. 9 and Supplementary Table 16). The NonCog PGS, independent of the Cog PGS, predicted observed variation in non-cognitive skills at all developmental stages. Associations were small at earlier ages (for example, $\beta = 0.07$, 95% CI 0.03 to 0.11, $P$ (corrected) = $1.93 \times 10^{-3}$) for parent-reported education-specific non-cognitive skills at 9 years and $\beta = 0.10$, 95% CI 0.08 to 0.12, $P$ (corrected) = $2.24 \times 10^{-11}$ for parent-reported self-regulation at 7 years) but they increased developmentally, particularly for self-reported education-specific non-cognitive measures ($\beta = 0.16$, 95% CI 0.12 to 0.20, $P$ (corrected) = $8.30 \times 10^{-17}$ at 16 years of age). The only exception was observed for self-reported education-specific non-cognitive skills at 9 years, for which the prediction was negative ($\beta = -0.03$, 95% CI −0.07 to 0.01) and did not reach significance after accounting for multiple testing (Supplementary Table 16).

In Supplementary Note 3a, we show that this increase in prediction was significant over time for the NonCog PGS, but not for the Cog PGS. Furthermore, we show that this increase is not explained by the NonCog PGS capturing more Cog variance later in adolescence (Supplementary Note 3b) or by SES (Supplementary Note 3c).

Cog and NonCog PGSs predicted variation in general cognitive ability, verbal ability and non-verbal ability at all developmental stages. As expected, the Cog PGS prediction of cognitive phenotypes was substantially stronger than the NonCog prediction, with estimates ranging between $\beta = 0.19$, 95% CI 0.17 to 0.21, $P$ (corrected) = $3.77 \times 10^{-42}$ and $\beta = 0.27$, 95% CI 0.23 to 0.31, $P$ (corrected) = $1.04 \times 10^{-52}$ for the Cog PGS and between $\beta = 0.10$, 95% CI 0.06 to 0.14, $P$ (corrected) = $4.41 \times 10^{-10}$ and $\beta = 0.18$, 95% CI 0.14 to 0.22, $P$ (corrected) $5.51 \times 10^{-21}$ for the NonCog PGS (Supplementary Fig. 10 and Supplementary Table 16).

Next, we considered the effects of the Cog and NonCog PGS on academic achievement over development. We detected associations between the Cog PGS and achievement as early as 7 years ($\beta = 0.24$, 95% CI 0.22 to 0.26, $P$ (corrected) = $3.68 \times 10^{-86}$), these associations remained largely consistent across development ($\beta = 0.26$, 95% CI 0.24 to 0.28, $P$ (corrected) = $2.71 \times 10^{-126}$ at 16 years of age). Although we observed weaker effects for the NonCog PGS in early childhood ($\beta = 0.10$, 95% CI 0.08 to 0.12, $P$ (corrected) = $8.12 \times 10^{-15}$) compared with the Cog PGS, these increased across development and reached effects comparable to those of the Cog PGS at 16 years ($\beta = 0.22$, 95% CI 0.20 to 0.24, $P$ (corrected) = $1.85 \times 10^{-84}$; Fig. 3b and Supplementary Table 16). The same pattern of associations was observed also when considering achievement in English and mathematics, separately (Supplementary Table 16). This observed increase in the NonCog PGS prediction of academic achievement over development is consistent with transactional models of gene–environment correlation ($r$GE), driven by NonCog genetics. These PGS predictions were in line with those obtained from the PGSs created using the GWAS-by-subtraction method published by Demange et al. (Supplementary Table 17).

## Within-family PGS–achievement associations

Given our observation of an increase in the NonCog PGS associations with academic achievement across development, we extended our pre-registered analyses (https://osf.io/m5f7j/) to examine whether and to what extent this increase was accounted for by family-wide processes. Specifically, using a sibling difference design, we separated the NonCog PGS associations into within-family effects, indexing direct genetic effects from between-family effects, which may include indirect genetic effects and demographic confounding (Methods). We examined Cog and NonCog within and between-family predictions of academic achievement from age 7 to 16 years.

Two main findings emerged from this analysis (Fig. 3c). First, we observed that the effect sizes for the direct effects of NonCog were about half the size of the population-level associations (Supplementary Table 18). Similarly, the prediction from the Cog PGS was reduced by over one-third, consistent with previous evidence[43]. Second, while the Cog direct and indirect genetic effects did not vary substantially over the developmental period considered (from $\beta = 0.20$, 95% CI 0.16 to 0.24, $P = 2.75 \times 10^{-20}$ to $\beta = 0.23$, 95% CI 0.19 to 0.27, $P = 4.12 \times 10^{-32}$), NonCog effects showed an increase from age 7 to age 16 (from $\beta = 0.06$, 95% CI 0.02 to 0.10, $P = 0.005$ to $\beta = 0.15$, 95% CI 0.11 to 0.19, $P = 1.39 \times 10^{-14}$; Fig. 3c and Supplementary Table 18). These results suggested that the developmental increase in the between-family PGS prediction was mostly driven by NonCog rather than Cog skills. In addition, this

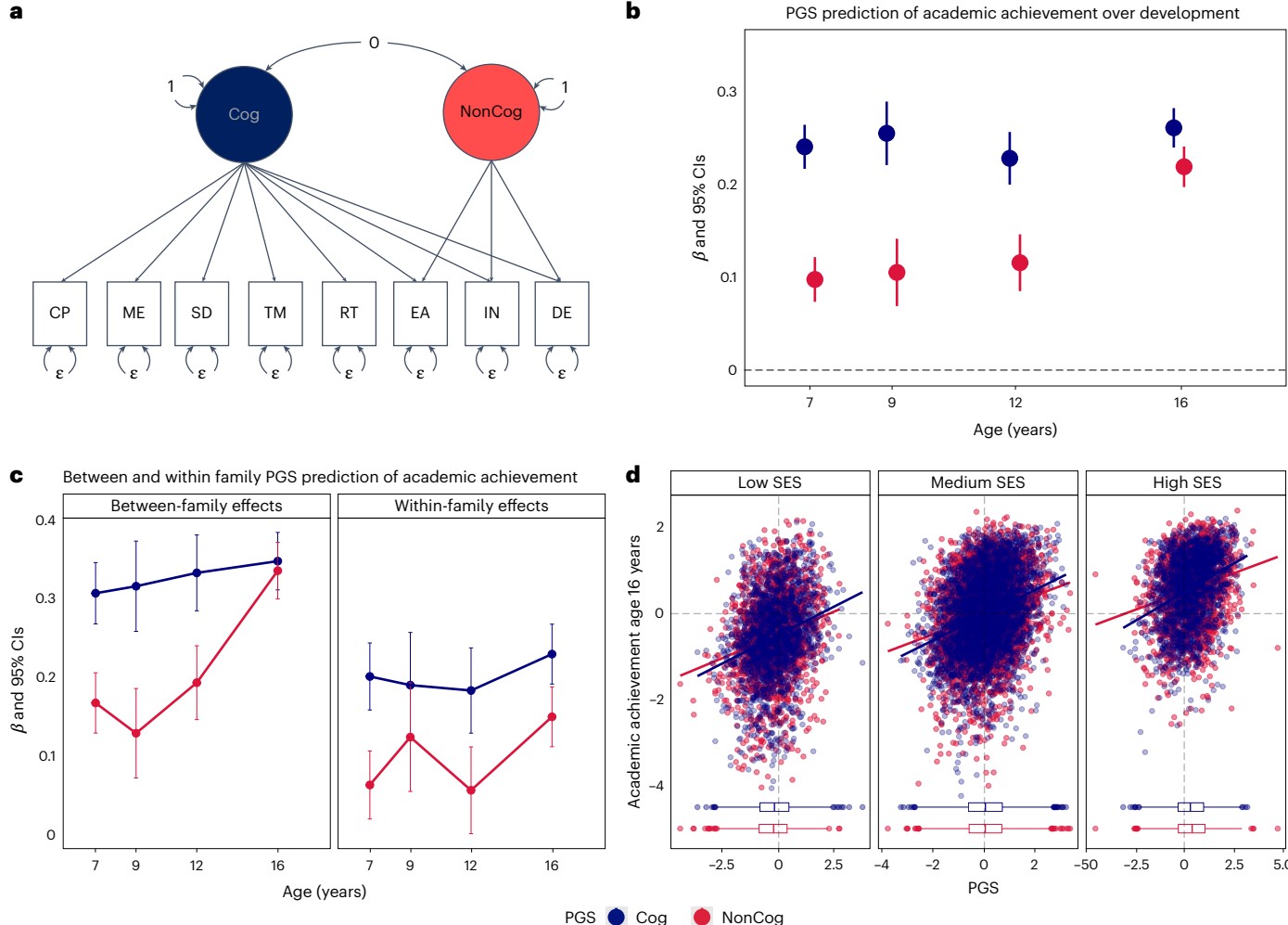

**Fig. 3 | Contribution of NonCog genetics to academic development: genomic analyses and gene–environment interplay. a**, Path diagram for the extension of the GWAS-by-subtraction model implemented in genomic SEM (Methods). In addition to GWAS summary statistics for cognitive performance (CP) and educational attainment (EA), summary statistics of memory (ME), symbol digit (SD), trail making (TM) and reaction time (RT) GWASs loaded on the Cog latent factor while GWAS summary statistics for income (IN) and deprivation (DE) loaded on the NonCog latent factor, in addition to EA (Methods). **b**, Cog (blue) and NonCog (red) polygenic score (PGS) prediction of academic achievement at ages 7, 9, 12 and 16 years. Point estimates represent beta coefficients and error bars are 95% CI, N of clustered observations 6,575 at age 7, 3,144 at age 9, 4,445 at age 12 and 7,307 at age 16 years. Each

cluster comprised twin siblings and non-independence was accounted for using generalized estimating equation. **c**, The results of PGS analyses after partitioning the effects of Cog and NonCog into between and within-family factors. Point estimates represent beta coefficients and error bars are 95% CI. **d**, Cog and NonCog PGS prediction of academic achievement at the end of compulsory education (age 16 years), plotted at different levels of family socio-economic status (SES): low (<25th percentile), middle (middle 50% interquartile range) and high (>75% percentile). The box plots show median interquartile range (middle 50%), minimum, maximum and outliers for Cog (blue) and NonCog (red), respectively. N ranging from 3,001 to 7,019, exact N for each regression analysis are reported in Supplementary Table 19.

developmental increase could be observed for both indirect and direct genetic effects. We conducted sensitivity analyses and replicated the results with the PGSs constructed using the method published by Demange et al. (Supplementary Table 18b).

**PGS–SES interactions on achievement**

Last, we extended our pre-registered analyses to test whether SES could explain or modify the observed pattern of developmental associations between PGS and academic achievement. We fitted multivariable models at each developmental stage, including Cog/NonCog PGS effects, along with SES at recruitment, covariates and their two-way interactions (Methods) to test whether SES moderated Cog and NonCog PGS effects on academic achievement. After adjusting for SES, the same pattern of relationships was observed, with a relatively stable association between the Cog PGS and achievement, and a steeper

increase in the NonCog PGS prediction, even though all effects were attenuated (Supplementary Table 19). We did not detect significant interaction effects between either the Cog or the NonCog PGS and SES (Supplementary Table 19).

Figure 3d depicts mutually adjusted slopes for the Cog and NonCog PGS prediction against academic achievement at different levels of family SES. The figure shows that although higher SES corresponded to greater achievement on average, the slope of the association between the Cog and NonCog PGS and achievement did not differ across socio-economic strata. Higher PGS, for both cognitive and non-cognitive skills, corresponded to higher academic achievement, and higher SES corresponded to both higher mean PGSs and higher achievement, indicating a correlation rather than an interaction between genetic and environmental influences on academic achievement.

## Discussion

We investigated the association between cognitive and non-cognitive genetics and academic achievement during compulsory education in a UK-based sample. Four complementary findings emerged. First, non-cognitive skills increasingly predicted academic achievement over the school years, and these associations remained substantial even after accounting for cognitive skills. Second, the links between non-cognitive skills and academic achievement were mainly due to shared genetic factors, whose relative contribution also increased over the school years. For example, the NonCog PGS prediction of academic achievement nearly doubled over the school years while the Cog PGS prediction remained relatively stable. Third, the increasingly important role of non-cognitive genetics persisted even after accounting for family-fixed effects. Fourth, PGS associations with academic achievement over development did not differ across socio-economic contexts. Together, these findings highlight the important role that non-cognitive skills play during primary and secondary education and suggest that fostering such skills might provide an avenue for successful educational strategies and interventions.

The first set of interesting findings about development emerged from twin analyses of the covariance between non-cognitive traits and academic achievement. First, we found that genetic factors accounted for most of the observed correlations between non-cognitive skills and academic achievement at all developmental stages. Second, both phenotypic and genetic correlation values increased developmentally, particularly for self-reported measures of non-cognitive traits. Third, our twin analyses showed that genetic factors accounted for most of the correlations between non-cognitive skills and academic achievement after accounting for cognitive skills. Finally, this independent genetic contribution of non-cognitive skills to academic achievement increased developmentally. This increase was observed for both education-specific non-cognitive skills, where the measures included in the general factors changed developmentally, as well as for domain-general self-regulation skills, for which the same measures were collected at all developmental stages. Therefore, the observed developmental increase in phenotypic and genetic associations independent of cognitive skills is unlikely to be an artefact of inconsistencies in measurement but rather reflects the increasingly important role of non-cognitive skills across compulsory education.

A further aim of the current study was to better understand what was captured by the NonCog PGS constructed using GWAS-by-subtraction[20], particularly in relation to what other skills beyond cognitive ability propel students down different educational trajectories. Given the link between SES and academic achievement[44], we were specifically interested in whether the NonCog PGS also indexed socio-economic-related factors. To this end, we extended the GWAS-by-subtraction model in two directions. First, with the aim of making a more refined cognitive factor, we added summary statistics from several other GWASs of fluid intelligence. Second, we included GWASs of other traits known to associate with achievement beyond cognitive abilities, specifically targeting SES-related traits, such as household income and social deprivation, making the NonCog PGS factor more explicitly socio-economic relevant.

It should be highlighted that GWAS of SES-relevant measures may be more subject to socio-demographic confounds, such that estimates of SNP effects will also capture population stratification phenomena, such as geographic clustering[45]. This limitation is particularly relevant for the GWAS of social deprivation as the measure is an area-based score of social inequality. Interestingly, the results obtained from this model paralleled those we obtained when we applied the Cog and NonCog PGSs from the original GWAS-by-subtraction model, which only used EA to define the NonCog factor. This suggests that the PGS measure of non-cognitive skills from Demange et al. may have already captured some SES-related effects. Importantly, our employment of

a within-family comparison helped us to mitigate possible confounds associated with uncontrolled population stratification.

Paralleling our multivariate twin results, we observed that the effects of the prediction from NonCog PGS to academic achievement increased from childhood to adolescence, beyond the effects of the Cog PGS. A few explanations are possible for this finding. First, this could be attributable to $r$GE, which could be passive, evocative or active[29,46]. Another explanation could be that PGSs become increasingly predictive during development as our sample gets closer in age to the adult samples where GWAS effect sizes were estimated in the case of EA and CP[17]. However, it is worth noting that this increase in prediction was not observed for the Cog PGS, for which effects on academic achievement were mostly developmentally stable. Moreover, our triangulation of results across multiple methods (including phenotypic and twin analyses) adds support to our finding of these developmental differences between Cog and NonCog genetics.

We applied a within-sibling design[43] to test whether environmental variables that are shared by siblings and that potentially confound PGS associations could explain the observed increase in the predictive power of the NonCog PGS. While the contributions of both PGSs were attenuated within family, suggesting a substantial role for environmental confounds shared by family members, an increase in the contribution of NonCog PGS to academic achievement from age 7 to 16 years was still evident when comparing siblings. In contrast, the within-family contribution of the Cog PGS remained relatively stable. The increase in the NonCog PGS prediction at the within-family level is consistent with transactional processes driven by active or evocative $r$GE[30,46,47] for NonCog PGS. As children grow up, they actively evoke or shape their environmental experiences based in part on their genetic dispositions, and these experiences in turn contribute to their academic development. Our findings suggest that children's educational experiences are increasingly shaped by their propensity towards non-cognitive skills.

To delve deeper into the role of socio-economic factors, we tested whether SES could modify the relationship between Cog and NonCog PGSs and academic achievement over development. While we did not find evidence for interaction effects in this regard, the Cog and NonCog PGS were conditionally independent in a multivariable model including SES, further indicating that the genetics captured by the NonCog factor were at least partly independent of SES-related genetic and environmental effects.

One caveat of these gene–environment interaction analyses is that adjusting for a heritable covariate, such as SES, can yield biased estimates in multivariable models including PGS[48,49]. Future work is needed to determine whether this is the case, perhaps leveraging results of within-family GWAS to construct PGS for 'direct' effects within families[50]. This limitation also pertains to our within-sibling PGS analyses, as it might be difficult to separate direct and indirect effects using population-based GWAS effects as a starting point[51]. Follow-up of these analyses employing PGS for direct effects obtained from family-based GWAS will shed light on this potential limitation. It should also be acknowledged that indirect genetic effects on education might reflect social stratification across generations in addition to nurturing processes that operate within nuclear families[52]. A further caveat of the present work is that, while we investigated genetic effects on non-cognitive skills and their link with academic achievement across development, we did not investigate stability and change using longitudinal models. Future work explicitly investigating developmental change at the phenotypic[53], genetic[28] and genomic[40,54] level, for example, using latent growth models[55], will address further developmental questions related to the role of non-cognitive skills in academic development.

To conclude, our study provides an in-depth investigation of the role of non-cognitive genetics in academic development. Triangulating multiple genetic and genomic methods, we found consistent evidence for the increasingly important role that non-cognitive skills

play during compulsory education. Genetic dispositions towards non-cognitive skills become increasingly predictive of academic achievement and, by late adolescence, they explain as much variance in achievement as do genetic dispositions towards cognitive skills. Results from within-family and developmental analyses are consistent with theorized transactional processes of active/evocative *r*GE by which, as they grow up, children evoke and actively select academic environments that correlate with their genetic disposition towards non-cognitive skills[29,46]. Fostering non-cognitive skills might provide a successful avenue for educational interventions.

## Methods

### Sample
Participants are part of the Twins Early Development Study (TEDS), a longitudinal study of twins born in England and Wales between 1994 and 1996. The families in TEDS are representative of the British population for their cohort in terms of socio-economic distribution, ethnicity and parental occupation. In total, 10,000 families are still actively involved with the TEDS study over 20 years after the first data collection wave (see ref. 56 for additional information on the TEDS sample). The present study includes data collected in TEDS across multiple waves. Specifically, we analysed data collected over five waves, when the twins were 4, 7, 9, 12 and 16 years old. The sample size and composition differ between collection waves, numbers for all measures included in the study are reported in Supplementary Table 1a. Individuals with severe medical conditions were excluded from analyses. These conditions include detrimental prenatal and postnatal conditions, as well as other conditions that could seriously impact later development. In addition, twins with uncertain and unknown zygosity were excluded from the analyses. TEDS has ethical approval from the research ethics committee of Kings College London (references: PNM/09/10–104 and HR/DP-20/21–22060). Consent was obtained before data collection at every wave.

### Measures
Below we provide a brief description of all the measures included in the present study. Please refer to https://www.teds.ac.uk/datadictionary for detailed descriptions of each measure and information on the items included in each construct.

**Education-specific non-cognitive skills.** At 9 years of age, data on education-specific non-cognitive skills were collected from parents, teachers and self-reports from the twins. Measures of academic self-perceived ability[57], academic interest[57] and the Classroom Environment Questionnaire (CEQ[58]) were available from all raters. The CEQ included the following subscales rated by parents and twins: (1) CEQ classroom satisfaction scale, (2) CEQ educational opportunities scale and (3) CEQ adventures scales that assessed enjoyment of learning. Ratings on the CEQ classroom satisfaction scale were also provided by the teachers.

At 12 years of age, data on education-specific non-cognitive skills were collected from parents, teachers and self-reports. The following measures were collected: academic self-perceived ability[57], academic interest[57], the mathematics environment questionnaire[59] and the literacy environment questionnaire[60]. The questionnaires asked several questions related to literacy and mathematics, including items, such as 'reading is one of my favourite activities', 'when I read books, I learn a lot' and 'in school, how often do you do maths problems from text books?', all rated on a four-point Likert scale.

At 16 years of age, education-specific non-cognitive skills were assessed via self-reports provided by the twins. The battery of education-specific non-cognitive constructs included the following measures:

(a) The brief academic self-concept scale included ten items (adapted from ref. 61), such as 'I like having difficult work to do' and 'I am clever', rated on a five-point Likert scale.

(b) School engagement[62] includes five subscales: teacher–student relations, control and relevance of schoolwork, peer support for learning, future aspirations and goals and family support for learning. The school engagement scale includes items, such as 'I enjoy talking to the teachers at my school', 'I feel like I have a say about what happens to me at school', 'school is important for achieving my future goals' and 'when I have problems at school, my family/carer(s) are willing to help me', rated on a four-point Likert scale.

(c) Grit was assessed with eight items from the Short Grit Scale[63] asking the twins to report on their academic perseverance answering questions such as 'setbacks don't discourage me' and 'I am a hard worker', rated on a five-point Likert scale.

(d) Academic ambition[64] was measured with five items asking participants to rate statements, such as the following 'I am ambitious' and 'achieving something of lasting importance is the highest goal in life' on a five-point Likert scale.

(e) Time spent studying mathematics was assessed with three items asking participants how much time every week they spent in 'regular lessons in mathematics at school', 'out-of-school-time lessons in mathematics' and 'study or homework in mathematics by themselves'.

(f) Mathematics self-efficacy[65] was measured with eight items asking students how confident they felt about having to perform different mathematics tasks, for example: 'calculating how many square metres of tiles you need to cover a floor' and 'understanding graphs presented in newspapers', rated on a four-point Likert scale

(g) Mathematics interest[65] asked participants to respond to three questions related to interest in mathematics, including 'I do mathematics because I enjoy it' and 'I am interested in the things I learn in mathematics'.

(h) Curiosity was assessed with seven items[66] asking participants to rate statements, such as 'when I am actively interested in something, it takes a great deal to interrupt me' and 'everywhere I go, I am looking out for new things or experiences' on a seven-point Likert scale

(i) Attitudes towards school was measured using the Programme for International Student Assessment attitudes to school measure[65], which included four items, such as 'school has helped give me confidence to make decisions' and 'school has taught me things, which could be useful in a job' rated on a four-point Likert scale.

**Self-regulation.** Emotional and behavioural self-regulation was assessed at all ages using the Strengths and Difficulties Questionnaire (SDQ)[67]. Data on domain-general self-regulation skills were collected from parents, teachers and self-reported by the twins. The SDQ includes five subscales: hyperactivity, conduct problems, peer problems, emotional problems and pro-social behaviour. Composite scores for all subscales except pro-social behaviour were reversed so that higher scores indicated higher levels of domain-general self-regulation skills. At 7 years of age, domain-general self-regulation skills were rated by the parents; at 9 years and 12 years by the parents, teachers and self-reported by the twins; and at 16 years self-reported by the twins.

**Cognitive ability.** At 7 years of age, cognitive ability was measured using four tests that were administered over the telephone by trained research assistants. Two tests assessed verbal ability: a 13-item similarity test and 18-item vocabulary test, both derived from the Wechsler Intelligence Scale for Children (WISC-III)[68]. Non-verbal ability was measured using two tests: a nine-item conceptual groupings test[69] and a 21-item WISC picture completion test[68]. Verbal and non-verbal ability composites were created by taking the mean of the standardized test scores within each domain. A general cognitive ability (*g*) composite

was derived taking the mean of the two standardized verbal and two standardized non-verbal test scores.

At 9 years of age, cognitive ability was assessed using four tests that were administered as booklets sent to TEDS families by post. Verbal ability was measured using the first 20 items from WISC-III-PI words test[70] and the first 18 items from WISC-III-PI general knowledge test[70]. Non-verbal ability was assessed using the shapes test (CAT3 Figure Classification)[71] and the puzzle test (CAT3 Figure Analogies)[71]. Verbal and non-verbal ability composites were created by taking the mean of the standardized test scores within each domain. A g composite was derived taking the mean of the two standardized verbal and two standardized non-verbal test scores.

At 12 years of age, cognitive ability was measured using four tests that were administered online. Verbal ability was measured using the full versions of the verbal ability tests administered at 9 years: the full 30 items from WISC-III-PI words test[70] and 30 items from WISC-III-PI general knowledge test[70]. Non-verbal ability was measured with the 24-item pattern test (derived from the Raven's Standard Progressive Matrices)[72] and the 30-item picture completion test (WISC-III-UK)[68]. Verbal and non-verbal ability composites were created by taking the mean of the standardized test scores within each domain. A g composite was derived from the mean of the two standardized verbal and two standardized non-verbal test scores.

At 16 years of age, cognitive ability was assessed using a composite of one verbal and one non-verbal test administered online. Verbal ability was assessed using an adaptation of the Mill Hill Vocabulary test[73] and non-verbal ability was measured using an adapted version of the Raven's Standard Progressive Matrices test[72]. A g composite was derived taking the mean of the two standardized tests.

**Academic achievement.** At 7 years of age, academic achievement was measured with standardized teacher reports and consisted of standardized mean scores of students' achievements in English and mathematics, in line with the National Curriculum level. Performance in English was assessed in four domains: speaking, listening, reading and writing abilities. Performance in maths was assessed in three domains: applying mathematics, as well as knowledge about numbers, shapes, space and measures.

At 9 years of age, academic achievement was again assessed using teacher reports. The domains assessed were the same for English and mathematics (although on age-appropriate content). In addition, performance in science was assessed considering two key domains: scientific enquiry and knowledge and understanding of life processes, living things and physical processes.

At 12 years of age, academic achievement was assessed in the same way as at age 9, with two exceptions. Mathematics added a fourth domain, data handling, and science added a third domain, materials and their properties. These additions were in line with the changes made to the National Curriculum teacher ratings.

At 16 years of age, academic achievement was measured using the General Certificate of Secondary Education (GCSE) examination scores. The GCSE is the UK nationwide examination usually taken by 16 year olds at the end of compulsory secondary education[74]. Twins' GCSE scores were obtained via mailing examination results forms to the families shortly after completion of the GCSE exams by the twins. For the GCSE, students could choose from a wide range of subjects. In the current analyses the mean score of the three compulsory GCSE subjects: English language and/or English literature, mathematics and a science composite (a mean score of any of the scientific subjects taken, including physics, chemistry and biology).

**Family SES.** At first contact, parents of TEDS twins received a questionnaire by post, and were asked to provide information about their educational qualifications, employment and mothers' age at first birth. A SES composite was created by standardizing these three variables

and calculating their mean. The same measures, except for mother's age at first birth, were used to measure family SES at 7 years of age. At age 16 years, data on SES were collected using a web questionnaire, and a total score was calculated from the standardized mean of five items: household IN, mother's and father's highest qualifications and mother's and father's employment status.

**Genetic data.** Two different genotyping platforms were used because genotyping was undertaken in two separate waves, 5 years apart. Affymetrix GeneChip 6.0 SNP arrays were used to genotype 3,665 individuals. Additionally, 8,122 individuals (including 3,607 dizygotic (DZ) co-twin samples) were genotyped on Illumina HumanOmniExpressExome-8v1.2 arrays. Genotypes from a total of 10,346 samples (including 3,320 DZ twin pairs and 7,026 unrelated individuals) passed quality control, including 3,057 individuals genotyped on Affymetrix and 7,289 individuals genotyped on Illumina. The final data contained 7,363,646 genotyped or well-imputed SNPs. For additional information on the treatment of these samples see ref. [75].

### Analytic strategies

Our analyses fully adhered to our pre-registration, available at the following link: https://osf.io/m5f7j. Although we did not deviate from our pre-registered analyses, we extended them to include two further analyses. First, we explored within-family PGS associations to examine whether and to what extent the PGS prediction of academic achievement could be accounted for by family-wide processes. Second, we explored PGS × SES interactions in predicting academic achievement across compulsory education.

**Factor analysis, correlations and regressions.** Confirmatory factor analysis was employed to create latent dimensions of non-cognitive skills and general Cog ability at all ages using the lavaan package for R[76]. On the basis of well-established literature on general cognitive ability (g) and previous work in the TEDS sample[77], we constructed one factor for g at each developmental stage. Each g factor was created by taking the weighted loadings of two verbal and two non-verbal tests (Measures and Supplementary Table 6). Confirmatory factor analysis was also employed to construct dimensions of non-cognitive characteristics. On the basis of previous meta-analytic work on the non-cognitive characteristics that matter for educational outcomes[9,78], we embraced a theoretical distinction between education-specific non-cognitive skills (for example, motivations, attitudes and goals) and broader, more de-contextualized measures of self-regulation (for example, behavioural and emotional regulation), and created separate factors for (1) education-specific non-cognitive skills and (2) domain-general self-regulation skills separately for ages and raters, including all the measures available at each age for each rater (see Supplementary Tables 2 and 3 for factor loadings and model fit indices).

We applied phenotypic correlations to examine the associations between non-cognitive skills (both observed measures and factors) and general cognitive ability and academic achievement at each age. We applied multiple regressions to explore the associations between non-cognitive skills and academic achievement accounting for general cognitive ability. We applied Benjamini–Hochberg correction[79] to account for multiple testing.

**The twin method.** The twin method allows for the decomposition of individual differences in a trait into genetic and environmental sources of variance by capitalizing on the genetic relatedness between monozygotic (MZ) twins, who share 100% of their genetic makeup, and DZ twins, who share on average 50% of the genes that differ between individuals. The method is further grounded in the assumption that both types of twins who are raised in the same family share their rearing environments to approximately the same extent[80]. By comparing how similar MZ and DZ twins are for a given trait (intraclass correlations),

it is possible to estimate the relative contribution of genetic factors and environments to variation in that trait. Heritability, the amount of variance in a trait that can be attributed to genetic variance (A), can be roughly estimated as double the difference between the MZ and DZ twin intraclass correlations[80]. The ACE model further partitions the variance into shared environment (C), which describes the extent to which twins raised in the same family resemble each other beyond their shared genetic variance, and non-shared environment (E), which describes environmental variance that does not contribute to similarities between twin pairs (and also includes measurement error).

The twin method can be extended to the exploration of the covariance between two or more traits (multivariate genetic analysis). Multivariate genetic analysis allows for the decomposition of the covariance between multiple traits into genetic and environmental sources of variance, by modelling the cross-twin cross-trait covariances. Cross-twin cross-trait covariances describe the association between two variables, with twin one's score on variable one correlated with twin two's score on variable two, which are calculated separately for MZ and DZ twins. The examination of shared variance between traits can be further extended to test the aetiology of the variance that is common between traits and of the residual variance that is specific to individual traits. We conducted these analyses in OpenMx for R[81].

It is possible to apply SEM to decompose latent factors into A, C and E components, applying models, such as the common pathway model. The common pathway model is a multivariate genetic model in which the variance common to all measures included in the analysis can be reduced to a common latent factor, for which the A, C and E components are estimated. As well as estimating the aetiology of the common latent factor, the model allows for the estimation of the A, C and E components of the residual variance in each measure that is not captured by the latent construct[82]. We conducted these analyses in Mplus[83].

A further multivariate twin method, grounded in SEM, is the Cholesky decomposition, which examines the genetic and environmental underpinnings of the associations between multiple variables or latent factors. The Cholesky approach parses the genetic and environmental variation in each trait into that accounted for by traits previously entered into the model and the variance, which is unique to a newly entered trait. In our case, the Cholesky decomposition partitions the genetic and environmental variance that is common across cognitive, non-cognitive and achievement measures from the genetic and environmental variance that is common between non-cognitive skills and achievement, independently of that accounted for by general cognitive ability. Cholesky decompositions were conducted on latent dimensions of cognitive and non-cognitive skills and observed variation in academic achievement (Supplementary Tables 12 and 13). We conducted these analyses in Mplus[83].

**Genomic SEM.** Genomic SEM[40] is an approach to conduct multivariate genome-wide association analyses. On the basis of the principles of SEM widely used in twin analyses and integrated with linkage disequilibrium score regression[84], genomic SEM jointly analyses GWAS summary statistics for multiple traits to test hypotheses about the structure of the genetic covariance between traits. Here, we employed genomic SEM to create latent GWAS summary statistics for unmeasured traits on the basis of other traits for which GWAS summary statistics exist. Recent work applied a GWAS-by-subtraction approach[20] leveraging the GWASs of EA[17] and CP[17,85] to obtain a GWAS of non-cognitive skills. The GWAS-by-subtraction approach estimates, for each SNP, an effect on EA that is independent of that SNP's effect on CP (therefore, indexing residual 'non-cognitive' SNP effects). The model regresses the EA and CP summary statistics on two latent variables, Cog and NonCog. EA and CP are both regressed on the Cog latent variable and only EA is regressed on the NonCog latent factor. The Cog and NonCog factors are specified to be uncorrelated and residual covariances across factor indicators are set to zero. Cog and NonCog are then regressed on each SNP, iterating across all SNPs in the genome.

We extended the GWAS-by-subtraction with the aim of obtaining potentially more fine-grained Cog and NonCog factors. Specifically, the model was extended as follows: loading exclusively on the Cog factor: five UK Biobank Cog traits (CP[85], symbol digit (SD) substitution, memory, trail making test and RT)[41]. Loading on both the Cog and NonCog factors: EA[17], Townsend deprivation index[86] and income[42]. An additional difference from the original GWAS-by-subtraction is that we let residual variances vary freely (that is, we did not constrain them to 0; Fig. 3a and Supplementary Table 14).

**PGS analyses.** PGSs were calculated as the weighted sums of each individual's genotype across all SNPs, using LDpred weights[87]. LDpred is a Bayesian shrinkage method that corrects for local linkage disequilibrium (that is, correlations between SNPs) using information from a reference panel (we used the target sample (TEDS) limited to unrelated individuals) and a prior for the genetic architecture of the trait. We constructed PGS using an infinitesimal prior, that is, assuming that all SNPs are involved in the genetic architecture of the trait, as this has been found to perform well with highly polygenic traits, such as EA, and in line with the approach adopted by Demange et al.[20]. In regression analyses, as with Demange et al.[20], both the Cog and NonCog PGSs were included in multiple regressions together with the following covariates: age, sex, the first ten principal components of ancestry and genotyping chip and batch. We accounted for non-independence of observation using the generalized estimating equation (GEE) R package[88].

**Within and between-family analyses.** We conducted within-sibling analyses using DZ twins to estimate family-fixed effects of both Cog and NonCog PGS on achievement across development[43]. A mixed model was fit to the data, including a random intercept to adjust for family clustering, and two family-fixed effects in addition to covariates (age, sex, the first ten principal components of ancestry and genotyping chip and batch): a between-family effect, indexed by the mean family PGS (that is, the average of the DZ twins' PGS within a family), and a within-family effect, indexed by the difference between each twin's PGS from the family mean PGS. Analyses were repeated with the PGS from Demange et al.[20] as sensitivity analyses.

**Gene–environment interaction analyses.** We conducted gene–environment interaction analyses to test whether SES moderated the effects of the Cog and NonCog PGS prediction on academic achievement over development. We fit a linear mixed model including Cog and NonCog PGS (the extensions), SES and their two-way interactions after adjusting for covariates (as above) and two-way interactions between predictors and covariates, plus a random intercept to adjust for family clustering. We adjusted for multiple testing using the Benjamini–Hochberg false discovery rate method[79] for all PGS analyses, at an α level of 0.05.

### Reporting summary
Further information on research design is available in the Nature Portfolio Reporting Summary linked to this article.

## Data availability
Researchers can apply for access to the TEDS data through their data access mechanism (www.teds.ac.uk/researchers/teds-data-access-policy). Summary statistics for the extended Cog and NonCog factors can be found via figshare at https://figshare.com/s/25abf6cc-4ca207468c6c (ref. 89).

## Code availability
Code is available via GitHub at https://github.com/CoDEresearchlab/NoncognitiveGenetics.

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

## Acknowledgements

We gratefully acknowledge the ongoing contribution of the participants in the TEDS and their families. TEDS has been supported by a programme grant to R.P. from the UK Medical Research Council (MR/M021475/1 and previously G0901245), with additional support from the US National Institutes of Health (AG046938). M.M. is supported by a starting grant from the School of Biological and Behavioural Sciences at Queen Mary University of London. R.P. is supported by a Medical Research Council Professorship award (G19/2). A.D.G. was supported by NIH Grants R01MH120219 and RF1AG073593. J.-B.P. is funded by the European Research Council under the European Union's Horizon 2020 research and innovation programme (grant agreement No. 863981) and the UK Research and Innovation (UKRI) under the UK government's Horizon Europe funding guarantee (grant number 575067), attributed to J.-B.P. and supporting A.G.A. P.A.D. is supported by the European Union (grant agreement number 101045526). Views and opinions expressed are, however, those of the authors only and do not necessarily reflect those of the European Union or the European Research Council Executive Agency. Neither the European Union nor the granting authority can be held responsible for them. The funders had no role in study design, data collection and analysis, decision to publish or preparation of the manuscript.

## Author contributions

M.M., A.G.A. and R.P. conceived and designed the study.; M.M. and A.G.A. analysed the data with helpful contributions from M.G.N. and P.B.; M.M., A.G.A., K.P.H. and R.P. wrote the paper with helpful contributions from M.G.N., P.B., K.R., R.C., S.v.S., P.A.D., E.v.B., A.D.G., L.R., J.D.l.F., J.-B.P. and E.M.T.-D. All authors contributed to the interpretation of data, provided critical feedback on manuscript drafts and approved the final draft.

## Competing interests

The authors declare no competing interests.

## Additional information

**Correspondence and requests for materials** should be addressed to Margherita Malanchini or Andrea G. Allegrini.

¹School of Biological and Behavioural Sciences, Queen Mary University of London, London, UK. ²Social, Genetic and Developmental Psychiatry Centre, King's College London, London, UK. ³Department of Clinical, Educational and Health Psychology, University College London, London, UK. ⁴Department of Biological Psychology, Vrije Universiteit Amsterdam, Amsterdam, the Netherlands. ⁵Department of Economics, Universita' di Bologna, Bologna, Italy. ⁶Royal Holloway University of London, London, UK. ⁷PROMENTA Research Center, Department of Psychology, University of Oslo, Oslo, Norway. ⁸Department of Education, University of York, York, UK. ⁹Research Institute LEARN!, Vrije Universiteit Amsterdam, Amsterdam, the Netherlands. ¹⁰Mental Health, Amsterdam Public Health Research Institute, Amsterdam, the Netherlands. ¹¹Institute for Behavioral Genetics, University of Colorado Boulder, Boulder, CO, USA. ¹²Max Planck Research Group Biosocial–Biology, Social Disparities and Development, Max Planck Institute for Human Development, Berlin, Germany. ¹³Department of Psychology, The University of Texas at Austin, Austin, TX, USA. ¹⁴These authors contributed equally: Margherita Malanchini, Andrea G. Allegrini. ✉e-mail: m.malanchini@qmul.ac.uk; a.allegrini@ucl.ac.uk

# Reporting Summary

## Statistics

For all statistical analyses, confirm that the following items are present in the figure legend, table legend, main text, or Methods section.

| n/a | Confirmed | |
|---|---|---|
| ☐ | ☒ | The exact sample size (*n*) for each experimental group/condition, given as a discrete number and unit of measurement |
| ☐ | ☒ | A statement on whether measurements were taken from distinct samples or whether the same sample was measured repeatedly |
| ☐ | ☒ | The statistical test(s) used AND whether they are one- or two-sided<br>*Only common tests should be described solely by name; describe more complex techniques in the Methods section.* |
| ☐ | ☒ | A description of all covariates tested |
| ☐ | ☒ | A description of any assumptions or corrections, such as tests of normality and adjustment for multiple comparisons |
| ☐ | ☒ | A full description of the statistical parameters including central tendency (e.g. means) or other basic estimates (e.g. regression coefficient) AND variation (e.g. standard deviation) or associated estimates of uncertainty (e.g. confidence intervals) |
| ☐ | ☒ | For null hypothesis testing, the test statistic (e.g. $F$, $t$, $r$) with confidence intervals, effect sizes, degrees of freedom and $P$ value noted<br>*Give P values as exact values whenever suitable.* |
| ☒ | ☐ | For Bayesian analysis, information on the choice of priors and Markov chain Monte Carlo settings |
| ☐ | ☒ | For hierarchical and complex designs, identification of the appropriate level for tests and full reporting of outcomes |
| ☐ | ☒ | Estimates of effect sizes (e.g. Cohen's *d*, Pearson's *r*), indicating how they were calculated |

*Our web collection on statistics for biologists contains articles on many of the points above.*

## Software and code

Policy information about availability of computer code

| Data collection | Participants are part of the Twins Early Development Study (TEDS), a longitudinal study of twins born in England and Wales between 1994 and 1996. The present study includes data collected in TEDS across multiple waves. Specifically, we analyzed data collected over five waves, when the twins were 4, 7, 9, 12 and 16 years old. Data was collected using paper-pencil, telephone and online tests and questionnaires. In the paper we provide a brief description of all the measures included in the present study and how they were collected. However, please refer to https://www.teds.ac.uk/datadictionary for detailed descriptions of each measure and information on the items included in each construct. |
|---|---|
| Data analysis | We used the following software and packages in our data analyses: R package `lavaan version 0.6-18` (contruction of all latent factors); Mplus version 8 (twin modelling of latent variables); OpenMx version 3 in R (Twin modelling of observed otcomes); Genomic SEM in R (modelling of genomic data and new GWAS of noncognitiev skills). LDpred1 (contruction of polygenic scores). Packages gee and tidyverse in R (multiple regressions including polygenic scores accounting for relatedness in the sample). Code is available at https://github.com/CoDEresearchlab/NoncognitiveGenetics. |

For manuscripts utilizing custom algorithms or software that are central to the research but not yet described in published literature, software must be made available to editors and reviewers. We strongly encourage code deposition in a community repository (e.g. GitHub). See the Nature Portfolio guidelines for submitting code & software for further information.

# Data

Policy information about availability of data

All manuscripts must include a data availability statement. This statement should provide the following information, where applicable:

- Accession codes, unique identifiers, or web links for publicly available datasets
- A description of any restrictions on data availability
- For clinical datasets or third party data, please ensure that the statement adheres to our policy

Code is available at https://github.com/CoDEresearchlab/NoncognitiveGenetics. Researchers can apply for access to the Twins Early Development Study (TEDS) data through their data access mechanism (see www.teds.ac.uk/researchers/teds-data-access-policy). Summary statistics for the extended Cognitive and Noncognitive factors can be found at the following link: https://figshare.com/s/25abf6cc4ca207468c6c.

# Research involving human participants, their data, or biological material

Policy information about studies with human participants or human data. See also policy information about sex, gender (identity/presentation), and sexual orientation and race, ethnicity and racism.

| | |
|---|---|
| Reporting on sex and gender | we analyzed data collected over five waves, when the twins were 4, 7, 9, 12 and 16 years old. The sample size and composition differ between collection waves, numbers for all measures included in the study are reported in Supplementary Table 1. |
| Reporting on race, ethnicity, or other socially relevant groupings | The families in TEDS are representative of the British population for their cohort in terms of socio-economic distribution, ethnicity and parental occupation. |
| Population characteristics | Two different genotyping platforms were used because genotyping was undertaken in two separate waves, 5 years apart. AffymetrixGeneChip 6.0 SNP arrays were used to genotype 3,665 individuals. Additionally, 8,122 individuals (including 3,607 DZ co-twin samples) were genotyped on Illumina HumanOmniExpressExome-8v1.2 arrays. Genotypes from a total of 10,346 samples (including 3,320 DZ twin pairs and 7,026 unrelated individuals) passed quality control, including 3,057 individuals genotyped on Affymetrix and 7,289 individuals genotyped on Illumina. The final data contained 7,363,646 genotyped or well-imputed SNPs. For additional information on the treatment of these samples see76. |
| Recruitment | Participants are part of the Twins Early Development Study (TEDS), a longitudinal study of twins born in England and Wales between 1994 and 1996. The families in TEDS are representative of the British population for their cohort in terms of socio-economic distribution, ethnicity and parental occupation. Ten thousand families are still actively involved with the TEDS study over twenty years after the first data collection wave (see57 for additional information on the TEDS sample). The present study includes data collected in TEDS across multiple waves. Recruitment is described in the box below. |
| Ethics oversight | Ethical approval was granted by King's College London's ethics board. |

Note that full information on the approval of the study protocol must also be provided in the manuscript.

# Field-specific reporting

Please select the one below that is the best fit for your research. If you are not sure, read the appropriate sections before making your selection.

☐ Life sciences  ☒ Behavioural & social sciences  ☐ Ecological, evolutionary & environmental sciences

For a reference copy of the document with all sections, see nature.com/documents/nr-reporting-summary-flat.pdf

# Behavioural & social sciences study design

All studies must disclose on these points even when the disclosure is negative.

| | |
|---|---|
| Study description | This is a quantitative, developmental study leveraging data collected as part of the Twins early development Study over 9 years, when participants were ages 7,9, 12 and 16. |
| Research sample | Participants are part of the Twins Early Development Study (TEDS), a longitudinal study of twins born in England and Wales between 1994 and 1996. The families in TEDS are representative of the British population for their cohort in terms of socio-economic distribution, ethnicity and parental occupation. we analyzed data collected over five waves, when the twins were 4, 7, 9, 12 and 16 years old. Demographic characteristics are provided in table 1a. |
| Sampling strategy | Families in England and Wales with twins born between January 1994 and December 1996 were identified using electronic birth records and invited to join TEDS through the Office of National Statistics (ONS). According to data from the ONS, there would have been approximately 30,350 multiple births between 1994 and 1996 (Birth Characteristics, 2023; Vital Statistics in the UK, 2021). Of the families contacted, 16,810 parents expressed interest in registering their twins to be part of the study, with 13,759 consenting to |

take part during the first wave when twins were 18 months old. For detailed information on the TEDS sample see Lockhart et al. 2023 https://doi.org/10.1002/jcv2.12154

**Data collection**

Since first contact, data have been collected at 2, 3, 4, 7, 8, 9, 10, 12, 14, 16, 18, 21, and most recently, 26 years. However, budgetary restraints meant the complete sample was not invited to take part in every wave. At ages where the budget was more constrained, the older school cohorts, which contained the majority of TEDS participants, were prioritised. Parents, twins, and teachers have all provided data at various time points, creating a rich multi-informant database. Data linkage efforts, including education records and environmental measures linked using the unique postcodes associated with home addresses, have further extended the TEDS dataset. Twins' parents provided informed consent at each wave of study participation until age 16, after which twins were asked directly. Data have been collected in several forms including paper/pencil questionnaires, telephone and online platforms.

**Timing**

Core waves of assessment, in which the full contactable sample are invited, took place at ages 4, 7, 8, 12, 14, 16, 18, 21 and 26 years. TEDS is an ongoing study.

**Data exclusions**

Individuals with severe medical conditions were excluded from analyses. These conditions include detrimental prenatal and postnatal conditions, as well as other conditions that could seriously impact later development. In addition, twins with uncertain and unknown zygosity were excluded from the analyses.

**Non-participation**

As with any longitudinal cohort study, the TEDS sample has experienced attrition since first contact. However, over 10,000 families remain involved in the study, with more than 6000 families providing data at both ages 21 and 26, see Lockhart et al. 2023 for a detailed description. https://doi.org/10.1002/jcv2.12154

**Randomization**

Twin analyses compare similarities between monozygotic and dizygotic twins, therefore randomization was not part of these aanalyses.

# Reporting for specific materials, systems and methods

We require information from authors about some types of materials, experimental systems and methods used in many studies. Here, indicate whether each material, system or method listed is relevant to your study. If you are not sure if a list item applies to your research, read the appropriate section before selecting a response.

## Materials & experimental systems

| n/a | Involved in the study |
|---|---|
| ☒ | ☐ Antibodies |
| ☒ | ☐ Eukaryotic cell lines |
| ☒ | ☐ Palaeontology and archaeology |
| ☒ | ☐ Animals and other organisms |
| ☒ | ☐ Clinical data |
| ☒ | ☐ Dual use research of concern |
| ☒ | ☐ Plants |

## Methods

| n/a | Involved in the study |
|---|---|
| ☒ | ☐ ChIP-seq |
| ☒ | ☐ Flow cytometry |
| ☒ | ☐ MRI-based neuroimaging |

## Plants

**Seed stocks**

*Report on the source of all seed stocks or other plant material used. If applicable, state the seed stock centre and catalogue number. If plant specimens were collected from the field, describe the collection location, date and sampling procedures.*

**Novel plant genotypes**

*Describe the methods by which all novel plant genotypes were produced. This includes those generated by transgenic approaches, gene editing, chemical/radiation-based mutagenesis and hybridization. For transgenic lines, describe the transformation method, the number of independent lines analyzed and the generation upon which experiments were performed. For gene-edited lines, describe the editor used, the endogenous sequence targeted for editing, the targeting guide RNA sequence (if applicable) and how the editor was applied.*

**Authentication**

*Describe any authentication procedures for each seed stock used or novel genotype generated. Describe any experiments used to assess the effect of a mutation and, where applicable, how potential secondary effects (e.g. second site T-DNA insertions, mosiacism, off-target gene editing) were examined.*

