## [Peer Review File · Nature Human Behaviour]

Peer Review Information

Journal: Nature Human Behaviour

Manuscript Title: Genetic associations between noncognitive skills and academic achievement over development

Corresponding author name(s): Margherita Malanchini & Andrea G. Allegrini

Reviewer Comments & Decisions:

Decision Letter, initial version:
--

8th June 2023

Dear Dr Malanchini,

Thank you once again for your manuscript, entitled "Genetic contributions of noncognitive skills to academic development", and for your patience during the peer review process.

Your Article has now been evaluated by 3 referees. You will see from their comments copied below that, although they find your work of considerable potential interest, they have raised quite substantial concerns. In light of these comments, we cannot accept the manuscript for publication, but would be interested in considering a revised version if you are willing and able to fully address reviewer and editorial concerns.

We hope you will find the referees' comments useful as you decide how to proceed. If you wish to submit a substantially revised manuscript, please bear in mind that we will be reluctant to approach the referees again in the absence of major revisions. We are committed to providing a fair and constructive peer-review process. Do not hesitate to contact us if there are specific requests from the reviewers that you believe are technically impossible or unlikely to yield a meaningful outcome.

Based on these reviewers' reports, we ask that in your revision you pay more attention to the clarity and presentation of your results, including reporting full statistics in the main text whenever you report your results. This may require substantive rewriting and slower pace. In addition, you will see that Reviewer #3 poses a potential challenge to your interpretation of your results, SES potentially being an important contributor, and NonCog PGS simply capturing more variance associated with cognitive ability. We ask that you explore and/or rule out these and other alternatives by performing additional analyses.

Finally, your revised manuscript must comply fully with our editorial policies and formatting requirements. Failure to do so will result in your manuscript being returned to you, which will delay its consideration. To assist you in this process, I have attached a checklist that lists all of our

requirements. If you have any questions about any of our policies or formatting, please don't hesitate to contact me.

If you wish to submit a suitably revised manuscript, we would hope to receive it within 2 months. I would be grateful if you could contact us as soon as possible if you foresee difficulties with meeting this target resubmission date.

- Include a "Response to the editors and reviewers" document detailing, point-by-point, how you addressed each editor and referee comment. If no action was taken to address a point, you must provide a compelling argument. When formatting this document, please respond to each reviewer comment individually, including the full text of the reviewer comment verbatim followed by your response to the individual point. This response will be used by the editors to evaluate your revision and sent back to the reviewers along with the revised manuscript.
- Highlight all changes made to your manuscript or provide us with a version that tracks changes.

[REDACTED]

Thank you for the opportunity to review your work. Please do not hesitate to contact me if you have any questions or would like to discuss the required revisions further.

Sincerely,

[REDACTED]

Reviewer expertise:

Reviewer #1: behavioural genetics, neuroscience

Reviewer #2: sociogenomics, economics

Reviewer #3: intelligence, genetics

REVIEWER COMMENTS:

Reviewer #1:

Remarks to the Author:

This is a well-conducted and comprehensive study into the longitudinal relationship between cognitive

and non-cognitive skills and educational attainment and the role of genetics and environment therein. I have some recommendations for improvements:

The introduction could be better constructed. It reads a bit messy now with the four questions spread out across paragraphs. Perhaps before elaborating on the four questions, there could be a short paragraph stating the four questions and explaining shortly what binds them together before elaborating about every separate question. The way it's presented in the tweet thread from the first author (<https://twitter.com/MarghMalanchini/status/1650455777591668736>) is more coherent.

Abstract: "The PGS for noncognitive skills predicted academic achievement developmentally, with prediction nearly doubling by age 16, pointing to gene-environment correlation (rGE)." I'm not sure why the conclusion of this sentence is that it's rGE?

Page 3: "Therefore, genetic differences between siblings are thought to be free from demographic confounding factors". This should say "young siblings", as older siblings can experience active rGE after migrating out of their parent's house. Actually, young siblings could still experience some active or evocative rGE, but perhaps to a much lesser extent.

Page 4: "Adopting a multi-method, developmental approach, these analyses (see preregistration here: <https://osf.io/m5f7j/>) address four core research questions providing a detailed account of the processes through which cognitive and noncognitive skills are linked to the development of individual differences in academic achievement." This should have been on a new line in a new section/paragraph (instead of part of the paragraph about the fourth question).

Some suggested edits:

Page 2: "Second, do genetic dispositions towards noncognitive skills become increasingly important for academic achievement across development?" -> "Second, does the contribution of the genetics of noncognitive skills to academic achievement change across development?"

Page 3: "Third, to what extent developmental changes in genetic effects on academic achievement evidence passive or evocative/active rGE?" -> "Third, to what extent do developmental changes in genetic effects on academic achievement reflect passive and/or evocative/active rGE?"

Page 4: "For self-reported education-specific noncognitive skills, the effect size of the relative prediction of achievement increased from $\beta = .10$ at age 9 (when the effect size for the cognitive prediction was $\beta = .46$) to $\beta = .28$ at age 12 (when the cognitive prediction was $\beta = .36$) to $\beta = .58$ at age 16 (when the cognitive prediction was $\beta = .39$). A developmental increase was also observed for self-reported measures of domain-general self-regulation skills, for which the predictive power increased from $\beta = .11$ at age 9 to $\beta = .21$ at age 16 after accounting for general cognitive ability (Supplementary Table 8)."

Is this increase over time significant?

Page 6: "The heritabilities of latent noncognitive dimensions ranged between 74% for self-reported domain-general self-regulation skills at age 9 and 93% for parent-reported education-related noncognitive skills at age 9 (Supplementary Note 2, Supplementary Tables 10-11 and Supplementary Figure 8)."

A heritability of 93% is unusually high. Any idea what's behind this peculiar estimate?

Page 6: "We then investigated whether the observed genetic associations between latent dimensions

of noncognitive skills and academic achievement could be accounted for by genetic factors associated with cognitive skills. We investigated this question with a series of trivariate Cholesky decompositions (Methods) the results of which are presented in Figure 2B. Each bar in Figure 2B is the outcome of a different trivariate Cholesky decomposition examining the extent to which genetic effects associated with noncognitive skills accounted for genetic effects on academic achievement after controlling for genetic effects associated with cognitive skills at the same age. The length of each bar indicates the proportion of variance in academic achievement that is accounted for by genetic factors (i.e., heritability). The yellow shadings show that genetic effects associated with cognitive skills account for between 21% and 36% of the total variance in academic achievement. The orange shadings indicate that genetic effects associated with noncognitive skills account for between 0.1% and 32.5% of the variance in academic achievement, independent of cognitive skills. The red shadings show that between 5% and 37% of the variance of academic achievement is independent of genetic effects associated with cognitive and noncognitive skills.”

This part reads more like a Figure legend than a results section.

Page 6: “Standardized squared path estimates increased from 1% of the total variance in academic achievement at age 9 to 4% at age 12 and 12% at age 16 (Supplementary Tables 12 and 13).”

Is this increase significant?

There are some error bars missing in Supplementary Figure 10.

Page 8: “The NonCog PGS, independent of the cognitive PGS, predicted observed variation in noncognitive skills at all developmental stages. Associations were small at earlier ages (e.g., $\beta = 0.07$, $SE = 0.02$, $p(\text{corrected}) = 1.93E-03$) for parent-reported education-specific noncognitive skills at 9, and $\beta = 0.10$, $SE = 0.01$, $p(\text{corrected}) = 2.24E-11$ for parent-reported self-regulation at 7) but they increased developmentally, particularly for self-reported education-specific noncognitive measures ($\beta = 0.16$, $SE = 0.02$, $p(\text{corrected}) = 8.30E-17$ at age 16).”

Were these increases significant?

Reviewer #2:

Remarks to the Author:

This paper is a fine-scaled analysis of the role of non-cognitive skills on educational achievement through childhood. The authors employ a constellation of methods, including twin studies, genetic correlation estimation, and polygenic prediction to understand the dynamic role of noncognitive skills. They find that the relationship between their measures of non-cognitive skills and educational achievement grow throughout childhood even after accounting for the role of cognitive skills. Their results were robust in both between- and within-family designs.

This paper was full of very interesting results which will likely be useful contributions to the literature. Have a number of concerns about the manuscript, however, most of which related to how these results are presented. I describe my concerns in more detail below.

Major points:

1) My largest concern is likely a consequence of the scale of the project. The text is incredibly dense, and statistics and analyses are often reported with very little help directing the reader. I generally

found that in order to understand several parts of the the results section, I needed to be flipping back and forth between the Methods Section, the Supplementary Note, and the Supplementary Tables.

As an example, in reference to Figure 2, the authors state "The length of each bar indicates the proportion of variance in academic achievement that is accounted for by genetic factors (i.e., heritability)." I spent several minutes trying to figure out why the length of the bars did not correspond to the heritability estimates reported in Supplementary Tables 10 and 11, and I eventually realized that the length of the bars was not actually the heritability, parenthetical notwithstanding. I believe it is instead the correlation between the genetic factor for the associated latent trait and educational achievement. (Is that right?) Then in trying to sort out where the numbers come from that describe the shading, I sifted through the hundreds of statistics that were reported in Supplementary Tables 12 and 13, and was unable figure out what numbers corresponded to the length of each bar shaded. After a close read of the Methods section and Supplementary Note, I think I was able to figure it all out (is it the squared values on the bottom row of each matrix?), but it shouldn't require this much work to understand a main text figure. This is particularly a problem since the figure doesn't include confidence intervals, instead pointing people to a complicated set of tables that have no table notes. (And if I understand where the numbers come from for this figure, the CIs on the squared values aren't actually even reported in the Supplementary Tables). To be honest, even after trying to sort out all this information, I'm still not confident that I'm actually right about what was done and what is being reported.

In general, I found myself having these sorts of troubles throughout the manuscript, which is a shame since there is so much interesting work in the paper, but it would be difficult to find it unless one is a very close reader. As a reviewer, it made it incredibly hard to assess the paper.

2) Relatedly, the Supplementary Tables would greatly benefit from Table Notes. A lot of them have key variables that are not defined. For example, what are A1, A2, and A3 in Tables 12 and 13. Or what is "ases" in Table 19.

3) In the main text, it would be helpful if any statistic that is reported also included a standard error of confidence interval. This is especially important for statistics corresponding to the main conclusions of the paper. For example, the authors state, "As expected, the results differed most for socioeconomic traits, with stronger correlations for NonCog than Cog with longevity ($r = 0.52$ Vs. $r = 0.34$), neighbourhood deprivation ($r = -0.66$, Vs. $r = -0.28$), and educational attainment $r = 0.83$ Vs. $r = 0.65$; Supplementary Figure 10 and Supplementary Table 15)." Looking at Supplementary Table 15, these differences are all obviously strongly significant, but you wouldn't know that from the main text.

Minor points:

4) The authors state that "The heritabilities of latent noncognitive dimensions ranged between 74% for self-reported domain-general self-regulation skills at age 9 and 93% for parent-reported education-related noncognitive skills at age 9 (Supplementary Note 2, Supplementary Tables 10-11 and Supplementary Figure 8)." This may be an error or it may be confusion on what is being reported, but I believe you estimate a 70% h^2 for self-report, education-specific, noncognitive skills at age 9, which is smaller than their lower bound of 74%.

5) The authors find a genetic correlation between Cog and NonCog of $r_g = 0.15$." I'm confused about why this would not be zero. Isn't NonCog defined such that it is genetically uncorrelated with Cog? If it

is as high as .15, does that raise questions about the interpretation of the rest of the results?

6) One thing that may explain the rising predictive power of the PGS over time is imperfect genetic correlation over time. Since the nonCog is based on an adult measure of educational attainment or SES, presumably, the nonCog component captured by the PGI more closely approximates nonCog at later ages. So even if noncognitive skills played an equally important (but heterogeneous) role across the life stage, you might still see the pattern observed in the PGS analysis. Is it possible to estimate the genetic correlation of cognitive and noncognitive factors across ages?

7) Main text Fig 3c needs some indication of what the dependent variable is. Presumably educational achievement, but it's not in the figure note or figure title.

Reviewer #3:

Remarks to the Author:

Malanchini et al. examine the role of non-cognitive skills in academic development using twin and genome-wide association study (GWAS) data. They show that across the ages of 9, 12, and 16 non-cognitive skills explain an increasing proportion of variance in academic achievement and that this is independent from cognitive ability. Next using GWAS and the GWAS by subtraction method Malanchini et al. derive individual SNP effects for cognitive ability and non-cognitive skills so that polygenic scores can be used to predict phenotypic variance in academic achievement. This is again done within the ages of 7, 9, 12, and 16 but also between and within families, and in high, medium and low socioeconomic status.

This is a well-designed project that is also well executed and I recommended publication. My points of criticism are minor but do influence the interpretation of the results made by the authors.

1. On page 9 the authors state that compared to their Cog PGS the effects of the NonCog PGS increased across development (Figure 3b). Furthermore, they claim these associations are also present when considering English and mathematics separately (Supplementary Table 16) and use this increase as evidence of a transactional model of gene-environment correlation. However, Figure 3b does not show an increase across development only that at age 16 the effect of the NonCog PGS is greater than it is for ages 7, 9, and 12. Upon consulting Supplementary Table 16 row 238 to row 249 I see that for English and mathematics there is no increase in the prediction of the NonCog PGS until the age of 16.

Have the authors considered that the NonCog PGS is simply capturing more variance associated with cognitive ability rather than noncognitive skills? We read that the phenotypic correlation between the latent measure of cognitive and non-cognitive skills increases with age (Supplementary Table 4), and that the genetic correlation between childhood IQ (range 6 – 18 years of age) and non-cognitive skills is $r_g = 0.45$ (Supplementary Table 15). Furthermore, Supplementary Table 16 shows that the NonCog PGS has the greatest point estimate for prediction at age 16 when compared to younger ages so it would seem plausible.

2. Similar claims are made with the within vs. between family comparisons. In the between family analysis there doesn't seem to be a consistent change over time only that the age 16 estimate is far higher than at earlier ages. In the within family analysis it is at best ambiguous as to whether there is any increase at from age 7 to age 16. Importantly, Supplementary Table 16 shows that the NonCog

PGS predicts the most variance in family SES, more so than the achievement variable described. Taking this information together it would seem that NonCog PGS best captures family SES (Supplementary Table 16 rows 305 and 306 show that the highest r^2 for the NonCog PGS are with family SES) and family SES is the variable responsible for the increase in the predictive power of the NonCog PGS at age 16. When differences the effects of family SES are removed as is the case for the within family analysis in Figure 3C. this increase in the predictive power of the NonCog PGS is also removed. If the NonCog PGS is simply providing a measure of the importance of family SES to achieving after the effects of the NonCog are controlled for this would change the conclusions of the paper.

Author Rebuttal to Initial comments

Response to Reviewers' comments.

Reviewer #1:

Remarks to the Author:

This is a well-conducted and comprehensive study into the longitudinal relationship between cognitive and non-cognitive skills and educational attainment and the role of genetics and environment therein.

Response: We thank the reviewer for their positive assessment of our study.

I have some recommendations for improvements:

The introduction could be better constructed. It reads a bit messy now with the four questions spread out across paragraphs. Perhaps before elaborating on the four questions, there could be a short paragraph stating the four questions and explaining shortly what binds them together before elaborating about every separate question. The way it's presented in the tweet thread from the first author (<https://twitter.com/MarghMalanchini/status/1650455777591668736>) is more coherent.

Response: We thank the reviewer for this suggestion, and we have modified the Introduction accordingly (see excerpt below)

“The current study uses both twin and DNA-based methods to expand our understanding of the role of noncognitive skills in academic development. We address four key questions (Figure 1). First, does the contribution of noncognitive skills to academic achievement change over development (from age 7 to age 16)? Second, do genetic predispositions to noncognitive skills vary in their contributions to academic achievement across development? Third, to what extent are these associations accounted for by between-family processes, such as environmental influences shared between individuals in a family? Fourth, do genetic contributions to academic achievement vary by socioeconomic status?” [Page 2 of the revised manuscript].

Abstract: “The PGS for noncognitive skills predicted academic achievement developmentally, with prediction nearly doubling by age 16, pointing to gene-environment correlation (rGE).”

I’m not sure why the conclusion of this sentence is that it’s rGE?

Response: We thank the reviewer for pointing this out. We have removed this portion of the sentence from the abstract. Additionally, in the Introduction and Discussion we now describe how active/evocative gene-environment correlations are one theorized mechanism for increases in the magnitude of genetic effects. For instance, in the Introduction we now write:

“Third, with a sibling-difference design, we examined to what extent the developmental relationship between genetic propensity for noncognitive skills and academic achievement was accounted for by family-wide environmental processes. Sibling differences in genotypes are randomized by meiosis, such that siblings have an equal probability of inheriting any given parental allele. Therefore, within-sibling pair PGS associations are thought to be less confounded by environmental differences between nuclear families, including population stratification and indirect genetic effects²⁶. Indirect genetic effects refer to the effects of the non-transmitted parental genotypes on the offspring phenotype, potentially reflecting rearing environments, although they can also capture broader demographic phenomena, such as assortative mating²⁷. Conversely, differences between siblings in PGS associations are often referred to as “direct” genetic effects^{28,29} in that they are consistent with a causal effect of genetic variants within an individual on their phenotype. However, even direct genetic effects involve mediation through environmental processes. For example, children with a greater motivation towards academic achievement might actively select, modify, and create environmental experiences that foster further achievement, such as deciding to take advanced classes²⁹. That is, genetic differences between children can result in differential exposure to learning environments, which, in turn, can affect their academic achievement,³⁰. These active/evocative gene-environment correlations (rGE) amplify the effects of genetic difference

and are one theorized mechanism for increasing genetic effects over development^{31,32}.” [Page 3 of the revised manuscript]

Page 3: “Therefore, genetic differences between siblings are thought to be free from demographic confounding factors”. This should say “young siblings”, as older siblings can experience active rGE after migrating out of their parent’s house. Actually, young siblings could still experience some active or evocative rGE, but perhaps to a much lesser extent.

Response: We appreciate the opportunity to clarify. We have now re-written this section to be clearer about what environmental factors are controlled for in the sibling comparison. Please see the paragraph quoted in the above response. Specifically, the sibling comparison controls for environmental processes shared by siblings raised in the same family, which are orthogonal to the randomization of genotypes within the family. But it does control for active/evocative rGE, which has been theorized to begin even in infancy and increase over development. Importantly, we do not consider active/evocative rGE to be a “confound” per se, but a hypothetical mediator of the direct genetic effects that are identified in a within-sibling analysis.

Page 4: “Adopting a multi-method, developmental approach, these analyses (see preregistration here: <https://osf.io/m5f7j/>) address four core research questions providing a detailed account of the processes through which cognitive and noncognitive skills are linked to the development of individual differences in academic achievement.” This should have been on a new line in a new section/paragraph (instead of part of the paragraph about the fourth question).

Response: We have changed it and we have added a sentence about triangulating evidence across different genetic methods. [Page 4 of the revised manuscript]

Some suggested edits:

Page 2: “Second, do genetic dispositions towards noncognitive skills become increasingly important for academic achievement across development?” -> “Second, does the contribution of the genetics of noncognitive skills to academic achievement change across development?”

Response: Thank you. We have changed this as suggested.

Page 3: “Third, to what extent developmental changes in genetic effects on academic achievement evidence passive or evocative/active rGE?” -> “Third, to what extent do

developmental changes in genetic effects on academic achievement reflect passive and/or evocative/active rGE?”

Response: We have changed this to read as follows: Third, to what extent are these associations accounted for by between-family processes, such as environmental influences shared between individuals in a family? [Page 2 of the revised manuscript]

Page 4: “For self-reported education-specific noncognitive skills, the effect size of the relative prediction of achievement increased from $\beta = .10$ at age 9 (when the effect size for the cognitive prediction was $\beta = .46$) to $\beta = .28$ at age 12 (when the cognitive prediction was $\beta = .36$) to $\beta = .58$ at age 16 (when the cognitive prediction was $\beta = .39$). A developmental increase was also observed for self-reported measures of domain-general self-regulation skills, for which the predictive power increased from $\beta = .11$ at age 9 to $\beta = .21$ at age 16 after accounting for general cognitive ability (Supplementary Table 8).”

Is this increase over time significant?

Response: Thank you for pointing this out. The increase over time is significant, as indicated by non-overlapping 95% confidence intervals. We now report 95% confidence intervals with all the estimates (see excerpt below).

“Latent factors of education-specific noncognitive skills and domain-general self-regulation skills were correlated positively with academic achievement at all developmental stages. Effect sizes differed by rater and developmental stage and tended to increase with age. For example, the association between self-rated education-specific noncognitive skills and academic achievement increased from $r = 0.10$ (95% CIs = 0.07; 0.14) at age 9, to $r = 0.41$ (95% CIs = 0.38; 0.44) at age 12, to $r = 0.51$ (95% CIs = 0.48; 0.55) at age 16 (see Supplementary Note 1, Supplementary Figure 2 and Supplementary Table 5). Latent noncognitive factors were also modestly correlated with latent factors of general cognitive ability (Supplementary Table 6) at the same age (Supplementary Table 7).” [Page 4 of the revised manuscript]

Page 6: “The heritabilities of latent noncognitive dimensions ranged between 74% for self-reported domain-general self-regulation skills at age 9 and 93% for parent-reported education-related noncognitive skills at age 9 (Supplementary Note 2, Supplementary Tables 10-11 and

Supplementary Figure 8).”

A heritability of 93% is unusually high. Any idea what’s behind this peculiar estimate?

Response: We welcome the Reviewer’s point about how the heritability estimates that we obtained for our latent dimensions of noncognitive skills are substantial. While this would be an unusual finding for observed self-reported noncognitive measures, this is not uncommon for latent dimensions, which exclude error of measurement. For example, Tucker-Drob et al. (2016) found a heritability of 0.69 (69%) for a general factor of noncognitive skills, called a general factor of character in the original paper, measured in childhood (Tucker-Drob et al., 2016). This estimate is in line with what we obtained for our latent self-reported measures of education-specific noncognitive skills: $h^2 = 0.70$ (95% CIs = 0.63; 0.78) at age 9, $h^2 = 0.75$ (95% CIs = 0.68; 0.82) at age 12, and $h^2 = 0.77$ (95% CIs = 0.74; 0.81) and of self-regulation.

We obtained higher heritability estimates for latent dimensions of noncognitive skills rated by parents and teachers. For example, individual differences in the latent factor of self-regulation constructed from parent reports collected when the twins were 7 years old were 86% heritable (95% CIs = 0.782; 0.940), similar estimates were obtained for teacher reports of self-regulation at age 9 ($h^2 = 0.87$; 95% CIs = 0.83; 0.90) and for education-specific noncognitive skills. Obtaining higher heritability estimates when considering measures rated by parents and teachers if compared to self-reports is also in line with what has been observed in previous studies. For example, Allegrini et al. 2020 estimated the heritability of a common factor of psychopathology (p factor) in childhood and found that the teacher-rated p factor was substantially more heritable (e.g., 80%; 95% CIs = .75; .81 at age 7) than a p factor constructed from self-reports (e.g., 54%; 95% CIs = .40; .68 at age 9 and 48%; 95% CIs = .36; .58 at age 12; Allegrini et al., 2020). We include a brief discussion of this issue in the revised manuscript:

“The heritabilities of latent noncognitive factors, which exclude error of measurement, ranged between 70% (95% CIs = 0.63; 0.77) for self-reported education-specific skills at age 9 and 93% (95% CIs = 0.91; 0.96) for parent-reported education-specific noncognitive skills at age 9 (Supplementary Note 2, Supplementary Tables 10-11 and Supplementary Figure 8). These substantial heritability estimates are consistent with previous studies that investigated the heritability of latent dimensions of noncognitive skills¹¹ and of a general factor of psychopathology across different raters³⁹.” [Page 6 of the revised manuscript]

Page 6: “We then investigated whether the observed genetic associations between latent dimensions of noncognitive skills and academic achievement could be accounted for by genetic

factors associated with cognitive skills. We investigated this question with a series of trivariate Cholesky decompositions (Methods) the results of which are presented in Figure 2B. Each bar in Figure 2B is the outcome of a different trivariate Cholesky decomposition examining the extent to which genetic effects associated with noncognitive skills accounted for genetic effects on academic achievement after controlling for genetic effects associated with cognitive skills at the same age. The length of each bar indicates the proportion of variance in academic achievement that is accounted for by genetic factors (i.e., heritability). The yellow shadings show that genetic effects associated with cognitive skills account for between 21% and 36% of the total variance in academic achievement. The orange shadings indicate that genetic effects associated with noncognitive skills account for between 0.1% and 32.5% of the variance in academic achievement, independent of cognitive skills. The red shadings show that between 5% and 37% of the variance of academic achievement is independent of genetic effects associated with cognitive and noncognitive skills.”

This part reads more like a Figure legend than a results section.

Response: We have now changed this section in line with Reviewer 1’s and Reviewer 2’s suggestions. We have modified the text, added a more detailed explanation of the method in the Results section and added confidence intervals for all the estimates reported (see excerpt below).

“We then investigated whether the observed genetic associations between latent noncognitive factors and academic achievement could be accounted for by genetic factors associated with cognitive skills. We investigated this question with a series of trivariate Cholesky decompositions (Methods) the results of which are presented in Figure 2B, which reports standardized squared path estimates, and Supplementary Tables 12 and 13, which report standardized path estimates and 95% confidence intervals. The Cholesky approach, similar to hierarchical regression, parses the genetic and environmental variation in each trait into that which is accounted for by traits that have been previously entered into the model and the variance which is unique to a newly entered trait.

Each bar in Figure 2B is the outcome of a different trivariate Cholesky decomposition of the heritability of academic achievement (the total length of the bar) into genetic effects associated with noncognitive skills after controlling for genetic effects associated with cognitive skills at the same age. We found that genetic effects associated with cognitive skills accounted for between 21% and 36% of the total variance in academic achievement, as indicated by standardized paths ranging between 0.46 (95% CIs = 0.37; 0.54) and 0.60 (95% CIs = 0.50; 0.70). Genetic effects associated with noncognitive skills, independent of cognitive skills, accounted for between 0.1% and 32.5% of the variance in academic achievement, independent

of cognitive skills, standardized paths ranged between 0.01 (95% CIs = -0.16; 0.17) for self-reported self-regulation at age 9 and 0.57 (95% CIs = 0.48; 0.67) for teacher reported education-specific noncognitive skills at age 9. Lastly, we found that between 5% and 37% of the variance in academic achievement was independent of genetic effects associated with cognitive and noncognitive skills; Standardized paths ranged between 0.23 (95% CIs = 0.13; 0.33) and 0.61 (0.52; 0.70).” [Page 6 of the revised manuscript]

Page 6: “Standardized squared path estimates increased from 1% of the total variance in academic achievement at age 9 to 4% at age 12 and 12% at age 16 (Supplementary Tables 12 and 13).” Is this increase significant?

Response: The increase was significant, particularly between self-reported noncognitive skills at ages 9 and 16, as indicated by non-overlapping 95% confidence intervals around the standardized path estimates. We have changed this section and now report 95% confidence intervals (see excerpt below).

“The top three rows of Figure 2B illustrate the developmental increase in how the genetics of self-reported noncognitive skills contribute to the genetics of academic achievement. Focusing on education-specific noncognitive skills, we found that standardized squared path estimates increased from 1% of the total variance in academic achievement at age 9 (standardized path estimate = 0.01 [95% CIs = -0.16; 0.17]) to 4% at age 12 (standardized path estimate = 0.16 [95% CIs = 0.02; 0.30]) and 12% at age 16 (standardized path estimate = 0.35 [95% CIs = 0.26; 0.44]) (Supplementary Tables 12 and 13). This increased contribution beyond cognitive skills was also observed for domain-general self-regulation. See Supplementary Figure 9 for the full models' results which include shared and nonshared environmental estimates.” [Page 6 of the revised manuscript]

There are some error bars missing in Supplementary Figure 10.

Response plan: Added

Page 8: “The NonCog PGS, independent of the cognitive PGS, predicted observed variation in noncognitive skills at all developmental stages. Associations were small at earlier ages (e.g., $\beta = 0.07$, $SE = 0.02$, $p(\text{corrected}) = 1.93E-03$) for parent-reported education-specific noncognitive

skills at 9, and $\beta = 0.10$, $SE = 0.01$, $p(\text{corrected}) = 2.24E-11$ for parent-reported self-regulation at 7) but they increased developmentally, particularly for self-reported education-specific noncognitive measures ($\beta = 0.16$, $SE = 0.02$, $p(\text{corrected}) = 8.30E-17$ at age 16).”
Were these increases significant?

Response: We have performed additional analyses in which we explicitly test whether the conditional estimates of Cog and NonCog PGS on academic achievement differ between measurement occasions (age 7 to 16), as it is now reported in Supplementary Note 3. Specifically, we fitted a series of Structural Equation Models where we fixed the parameter estimates of either the Cog, or the NonCog PGS on educational achievement (EA) to be equal across development (i.e., from ages 7 to 16). We then compared these models to a baseline model where we allowed all coefficients to freely vary (see Supplementary Note 3a for a schematic depiction of the model). A chi-square difference test of these nested models favored a model in which the effects of the Cog PGS on academic achievement across development were constrained to be equal, while the NonCog PGS effects were allowed to freely vary over time. [see Supplementary Note 3a for a description of the results]

Additional References:

Allegrini, A.G., Cheesman, R., Rimfeld, K., Selzam, S., Pingault, J.-B., Eley, T.C. and Plomin, R. (2020), *The p factor: genetic analyses support a general dimension of psychopathology in childhood and adolescence*. *J Child Psychol Psychiatr*, 61: 30-39. <https://doi.org/10.1111/jcpp.13113>

Tucker-Drob, E. M., Briley, D. A., Engelhardt, L. E., Mann, F. D., & Harden, K. P. (2016). *Genetically-mediated associations between measures of childhood character and academic achievement*. *Journal of Personality and Social Psychology*, 111(5), 790–815. <https://doi.org/10.1037/pspp0000098>

Reviewer #2:

Remarks to the Author:

This paper is a fine-scaled analysis of the role of non-cognitive skills on educational achievement through childhood. The authors employ a constellation of methods, including twin studies, genetic correlation estimation, and polygenic prediction to understand the dynamic role of noncognitive skills. They find that the relationship between their measures of non-cognitive skills and educational achievement grow throughout childhood even after accounting for the role of cognitive skills. Their results were robust in both between- and within-family designs.

This paper was full of very interesting results which will likely be useful contributions to the literature.

Response: We thank the reviewer for their positive assessment of our study.

Have a number of concerns about the manuscript, however, most of which related to how these results are presented. I describe my concerns in more detail below.

Major points:

1) My largest concern is likely a consequence of the scale of the project. The text is incredibly dense, and statistics and analyses are often reported with very little help directing the reader. I generally found that in order to understand several parts of the the results section, I needed to be flipping back and forth between the Methods Section, the Supplementary Note, and the Supplementary Tables.

Response: We have re-written the manuscript with the aim of improving its clarity, in response to Reviewer 1, we provide an example of how we have re-written part of the Results section, however, substantial changes to the writing have been made throughout the paper.

As an example, in reference to Figure 2, the authors state "The length of each bar indicates the proportion of variance in academic achievement that is accounted for by genetic factors (i.e., heritability)." I spent several minutes trying to figure out why the length of the bars did not correspond to the heritability estimates reported in Supplementary Tables 10 and 11, and I eventually realized that the length of the bars was not actually the heritability, parenthetical notwithstanding. I believe it is instead the correlation between the genetic factor for the associated latent trait and educational achievement. (Is that right?) Then in trying to sort out

where the numbers come from that describe the shading, I sifted through the hundreds of statistics that were reported in Supplementary Tables 12 and 13, and was unable figure out what numbers corresponded to the length of each bar shaded. After a close read of the Methods section and Supplementary Note, I think I was able to figure it all out (is it the squared values on the bottom row of each matrix?), but it shouldn't require this much work to understand a main text figure. This is particularly a problem since the figure doesn't include confidence intervals, instead pointing people to a complicated set of tables that have no table notes. (And if I understand where the numbers come from for this figure, the CIs on the squared values aren't actually even reported in the Supplementary Tables). To be honest, even after trying to sort out all this information, I'm still not confident that I'm actually right about what was done and what is being reported.

In general, I found myself having these sorts of troubles throughout the manuscript, which is a shame since there is so much interesting work in the paper, but it would be difficult to find it unless one is a very close reader. As a reviewer, it made it incredibly hard to assess the paper.

Response: We thank the reviewer for this comment, and we have now changed the Results section substantially. For example, we have rewritten the section describing the trivariate Cholesky decomposition, clarifying where all estimates came from and providing further information on standardized paths and related 95% confidence intervals (see Pages 7 and 8 of the revised manuscript).

More generally, throughout the Results section, we now report 95% confidence intervals or standard errors for all the estimates we describe. We have moved parts of our Methods section to the Results so that every analysis, particularly those less mainstream (e.g., GWAS-by-subtraction and polygenic scores on Page 9 of the revised manuscript and within family polygenic score analyses on Page 11-12), is briefly explained to a non-specialist audience before the Results are described.

2) Relatedly, the Supplementary Tables would greatly benefit from Table Notes. A lot of them have key variables that are not defined. For example, what are A1, A2, and A3 in Tables 12 and 13. Or what is "ases" in Table 19.

Response: We have added Table Notes and missing definitions to all our Supplementary Tables.

3) In the main text, it would be helpful if any statistic that is reported also included a standard error of confidence interval. This is especially important for statistics corresponding to the main conclusions of the paper. For example, the authors state, "As expected, the results differed most for socioeconomic traits, with stronger correlations for NonCog than Cog with longevity ($r = 0.52$ Vs. $r = 0.34$), neighbourhood deprivation ($r = -0.66$, Vs. $r = -0.28$), and educational attainment $r = 0.83$ Vs. $r = 0.65$; Supplementary Figure 10 and Supplementary Table 15)." Looking at Supplementary Table 15, these differences are all obviously strongly significant, but you wouldn't know that from the main text.

Response: Thank you for this suggestion, we have now added standard errors or 95% confidence intervals to all the statistics that we report in the manuscript.

Minor points:

4) The authors state that "The heritabilities of latent noncognitive dimensions ranged between 74% for self-reported domain-general self-regulation skills at age 9 and 93% for parent-reported education-related noncognitive skills at age 9 (Supplementary Note 2, Supplementary Tables 10-11 and Supplementary Figure 8)." This may be an error or it may be confusion on what is being reported, but I believe you estimate a 70% h^2 for self-report, education-specific, noncognitive skills at age 9, which is smaller than their lower bound of 74%.

Response: Thank you for spotting this error, we have now changed it.

5) The authors find a genetic correlation between Cog and NonCog of $r_g = 0.15$." I'm confused about why this would not be zero. Isn't NonCog defined such that it is genetically uncorrelated with Cog? If it is as high as .15, does that raise questions about the interpretation of the rest of the results?

Response: Even though by model specification Cog and NonCog covariance is set to 0, we note that residual variances of the indicators are not constrained to 0, contrary to the original GWAS by subtraction (Demange et al., 2021). Hence it may be that a degree of correlation in the exported genomic factors was reintroduced by means of these residual variances. However, we are reassured in the interpretation of the results by our sensitivity analyses using the Demange et al. summary statistics, which led to the same conclusions. Furthermore, both Cog and NonCog PGS are jointly fit in all our multivariable models, therefore effects reported are conditionally independent.

6) One thing that may explain the rising predictive power of the PGS over time is imperfect genetic correlation over time. Since the nonCog is based on an adult measure of educational attainment or SES, presumably, the nonCog component captured by the PGI more closely approximates nonCog at later ages. So even if noncognitive skills played an equally important (but heterogeneous) role across the life stage, you might still see the pattern observed in the PGS analysis. Is it possible to estimate the genetic correlation of cognitive and noncognitive factors across ages?

Response: Thank you for raising this issue. Although it is not possible at the moment to calculate the genetic correlation between cognitive and noncognitive genetics across ages using genomic methods, we calculated it using twin design. Overall, we found that genetic correlations with a factor of cognitive skills were weaker for measures of domain-general self-regulation if compared to education-specific noncognitive skills. For self-regulation measures, genetic correlations ranged between 0.16 and 0.37. For education-specific noncognitive measures, genetic correlations with general cognitive ability ranged between 0.25 and 0.64. Although there were some differences, these did not mirror the developmental pattern of increase in genetic effects that we found in our twin and PGS analyses. For example, the genetic correlation between domain-general self-regulation and cognitive skills was stable developmentally, and the strongest association observed between self-reported self-regulation and g at age 9 ($r_g = 0.45$). Similarly, the association between education-specific noncognitive skills and general cognitive ability was highly stable between ages 12 and 16 ($r_g = 0.64$ and 0.62 , respectively). A table including the genetic correlations is reported below.

Noncognitive factor	Age	Rater	g measure	Genetic correlation
Self-regulation	7	parent	G age 7	0.19
Self-regulation	9	parent	G age 9	0.37
Self-regulation	9	teacher	G age 9	0.25
Self-regulation	9	self	G age 9	0.45
Self-regulation	12	parent	G age 12	0.33
Self-regulation	12	teacher	G age 12	0.16
Self-regulation	12	self	G age 12	0.30
Self-regulation	16	self	G age 16	0.26
Education-specific	9	parent	G age 9	0.61
Education-specific	9	teacher	G age 9	0.59
Education-specific	9	self	G age 9	0.25

Education-specific	12	self	G age 12	0.64
Education-specific	16	self	G age 16	0.62

7) Main text Fig 3c needs some indication of what the dependent variable is. Presumably educational achievement, but it's not in the figure note or figure title.

Response: We have added a title to Figure 3C, which now reads:

'Between and within family PGS prediction of academic achievement'

Reviewer #3:

Remarks to the Author:

Malanchini et al. examine the role of non-cognitive skills in academic development using twin and genome-wide association study (GWAS) data. They show that across the ages of 9, 12, and 16 non-cognitive skills explain an increasing proportion of variance in academic achievement and that this is independent from cognitive ability. Next using GWAS and the GWAS by subtraction method Malanchini et al. derive individual SNP effects for cognitive ability and non-cognitive skills so that polygenic scores can be used to predict phenotypic variance in academic achievement. This is again done within the ages of 7, 9, 12, and 16 but also between and within families, and in high, medium and low socioeconomic status.

This is a well-designed project that is also well executed and I recommended publication. My points of criticism are minor but do influence the interpretation of the results made by the authors.

1. On page 9 the authors state that compared to their Cog PGS the effects of the NonCog PGS increased across development (Figure 3b). Furthermore, they claim these associations are also present when considering English and mathematics separately (Supplementary Table 16) and use this increase as evidence of a transactional model of gene-environment correlation.

However, Figure 3b does not show an increase across development only that at age 16 the effect of the NonCog PGS is greater than it is for ages 7, 9, and 12. Upon consulting Supplementary Table 16 row 238 to row 249 I see that for English and mathematics there is no increase in the prediction of the NonCog PGS until the age of 16.

Have the authors considered that the NonCog PGS is simply capturing more variance associated with cognitive ability rather than noncognitive skills? We read that the phenotypic correlation between the latent measure of cognitive and non-cognitive skills increases with age (Supplementary Table 4), and that the genetic correlation between childhood IQ (range 6 – 18

years of age) and non-cognitive skills is $rg = 0.45$ (Supplementary Table 15). Furthermore, Supplementary Table 16 shows that the NonCog PGS has the greatest point estimate for prediction at age 16 when compared to younger ages so it would seem plausible.

Response: We thank the reviewer for this comment. We have thought of different ways in which this issue could be addressed. First, we would like to refer Reviewer 3 to our response to Reviewer 2's point 6 above where we present the genetic correlations between noncognitive factors and general cognitive ability over development. These correlations were mostly stable developmentally, and particularly between ages 12 and 16, where we see the sharpest increase in the NonCog PGS prediction of academic achievement. These correlations suggest that noncognitive factors at age 16 do not capture more genetic variance that is shared with general cognitive ability if compared to younger ages.

We also thought of an alternative and potentially more direct way of testing this issue. We re-ran our Cog and NonCog PGS predictions of academic achievement also including phenotypic general cognitive ability (g) in our multivariate models. This very stringent test allowed us to examine whether the noncognitive PGS could still predict academic achievement at age 16 even after accounting for all the variance (not only genetic) shared with general cognitive ability. The results of our PGS analyses are reported in Supplementary Note 3b and in the table below. Although the effect of the prediction was attenuated, the NonCog PGS remained a significant predictor of variation in academic achievement at age 16.

Table S3.1: Noncognitive polygenic score prediction of academic achievement over development accounting for phenotypic general cognitive ability (g).

Measure of achievement	Measure of g	β	Robust standard error	p value
Achievement age 7	g age 7	0.06472661	0.01377481	2.62E-06
Achievement age 9	g age 9	0.08331642	0.01697248	9.16E-07
Achievement age 12	g age 12	0.07146178	0.01742392	4.11E-05
Achievement age 16	g age 16	0.1629107	0.01516482	6.42E-27

Note: All multivariate regression included the following predictors (in addition to the NonCog PGS and g): Cog PGS, the first 10 principal components of ancestry, genotyping chip. All variables were residualized for age and sex and the residuals were standardized before regression analyses.

We now include these additional analyses in the Supplementary Material, and we discuss them in the revised manuscript (see excerpt below).

“In Supplementary Note 3a, we show that this increase in prediction was significant overtime for the NonCog PGS, but not for the Cog PGS. Furthermore, we show that this increase was not explained by the NonCog PGS capturing more cognitive variance later in adolescence (Supplementary Note 3b). [Page 10 of the revised manuscript]

2. Similar claims are made with the within vs. between family comparisons. In the between family analysis there doesn't seem to be a consistent change over time only that the age 16 estimate is far higher than at earlier ages. In the within family analysis it is at best ambiguous as to whether there is any increase at from age 7 to age 16. Importantly, Supplementary Table 16 shows that the NonCog PGS predicts the most variance in family SES, more so than the achievement variable described. Taking this information together it would seem that NonCog PGS best captures family SES (Supplementary Table 16 rows 305 and 306 show that the highest r^2 for the NonCog PGS are with family SES) and family SES is the variable responsible for the increase in the predictive power of the NonCog PGS at age 16. When differences the effects of family SES are removed as is the case for the within family analysis in Figure 3C. this increase in the predictive power of the NonCog PGS is also removed. If the NonCog PGS is simply providing a measure of the importance of family SES to achieving after the effects of the NonCog are controlled for this would change the conclusions of the paper.

Response: Thank you for raising this point. To address the Reviewer's concern, we have expanded on our response to Reviewer 1's last comment and extended our analyses to explicitly test whether there is an increase in the association between Cog and NonCog PGS with academic achievement over time after adjusting for SES. These analyses indicated that even after adjusting for family SES, the developmental increase in the NonCog PGS prediction was significant, while that was not the case for the Cog PGS. These analyses are reported in Supplementary Note 3c and in the revised manuscript (see excerpt below).

“In Supplementary Note 3a, we show that this increase in prediction was significant overtime for the NonCog PGS, but not for the Cog PGS. Furthermore, we show that this increase was not explained by the NonCog PGS capturing more cognitive variance later in adolescence

(Supplementary note 3b), nor by socio-economic status (Supplementary Note 3c).” [Page 9 of the revised manuscript]

Decision Letter, first revision:

17th May 2024

Dear Dr. Malanchini,

Thank you for your patience as we've prepared the guidelines for final submission of your Nature Human Behaviour manuscript, "Genetic contributions of noncognitive skills to academic development" (NATHUMBEHAV-23041048A). Please see the following information, as well as the below message from the editor:

Dear Dr Malanchini,

I can't apologize enough once again for the extraordinary delay in sharing our final requests with you. I understand how frustrating the long wait has been and I am very sorry once again that the publication of your work has been delayed.

You can find attached our requests for your manuscript. We will make sure to prioritize the processing of your finalized manuscript to avoid further delays.

Thank you very much once again for your patience during this lengthy process.

With warm regards,

[REDACTED]

Please carefully follow the step-by-step instructions provided in the attached file, and add a response in each row of the table to indicate the changes that you have made. Please also address the additional marked-up edits we have proposed within the reporting summary. Ensuring that each point is addressed will help to ensure that your revised manuscript can be swiftly handed over to our production team.

We would hope to receive your revised paper, with all of the requested files and forms within two-three weeks. Please get in contact with us if you anticipate delays.

If you have not done so already, please alert us to any related manuscripts from your group that are under consideration or in press at other journals, or are being written up for submission to other journals (see: <https://www.nature.com/nature-research/editorial-policies/plagiarism#policy-on->

duplicate-publication for details).

Nature Human Behaviour offers a Transparent Peer Review option for new original research manuscripts submitted after December 1st, 2019. As part of this initiative, we encourage our authors to support increased transparency into the peer review process by agreeing to have the reviewer comments, author rebuttal letters, and editorial decision letters published as a Supplementary item. When you submit your final files please clearly state in your cover letter whether or not you would like to participate in this initiative. Please note that failure to state your preference will result in delays in accepting your manuscript for publication.

In recognition of the time and expertise our reviewers provide to Nature Human Behaviour's editorial process, we would like to formally acknowledge their contribution to the external peer review of your manuscript entitled "Genetic contributions of noncognitive skills to academic development". For those reviewers who give their assent, we will be publishing their names alongside the published article.

Cover suggestions

We welcome submissions of artwork for consideration for our cover. For more information, please see our guide for cover artwork.

ORCID

Non-corresponding authors do not have to link their ORCIDs but are encouraged to do so. Please note that it will not be possible to add/modify ORCIDs at proof. Thus, please let your co-authors know that if they wish to have their ORCID added to the paper they must follow the procedure described in the following link prior to acceptance: <https://www.springernature.com/gp/researchers/orcid/orcid-for-nature-research>

Nature Human Behaviour has now transitioned to a unified Rights Collection system which will allow our Author Services team to quickly and easily collect the rights and permissions required to publish your work. Approximately 10 days after your paper is formally accepted, you will receive an email in providing you with a link to complete the grant of rights. If your paper is eligible for Open Access, our Author Services team will also be in touch regarding any additional information that may be required to arrange payment for your article.

Please note that *Nature Human Behaviour* is a Transformative Journal (TJ). Authors may publish their research with us through the traditional subscription access route or make their paper immediately open access through payment of an article-processing charge (APC). Authors will not be required to make a final decision about access to their article until it has been accepted. Find out more about Transformative Journals

[REDACTED]

Best regards,
[REDACTED]

On behalf of

[REDACTED]

Reviewer #1:

Remarks to the Author:

I am happy with the way my comments were incorporated. I have one more comment:

It would be good if the authors incorporated this recent NHB article on indirect genetic effects in their discussion on their within- versus between-family results, since this article shows that much of what we think of as indirect genetic effects are actually from a wider familial dynastic source:

<https://www.nature.com/articles/s41562-023-01796-2>

Reviewer #3:

Remarks to the Author:

Reviewer #3:

Remarks to the Author:

Malanchini et al. examine the role of non-cognitive skills in academic development using twin and genome-wide association study (GWAS) data. They show that across the ages of 9, 12, and 16 non-cognitive skills explain an increasing proportion of variance in academic achievement and that this is independent from cognitive ability. Next using GWAS and the GWAS by subtraction method Malanchini et al. derive individual SNP effects for cognitive ability and non-cognitive skills so that polygenic scores can be used to predict phenotypic variance in academic achievement. This is again done within the ages of 7, 9, 12, and 16 but also between and within families, and in high, medium and low

socioeconomic status.

This is a well-designed project that is also well executed and I recommended publication. My points of criticism are minor but do influence the interpretation of the results made by the authors.

1. On page 9 the authors state that compared to their Cog PGS the effects of the NonCog PGS increased across development (Figure 3b). Furthermore, they claim these associations are also present when considering English and mathematics separately (Supplementary Table 16) and use this increase as evidence of a transactional model of gene-environment correlation. However, Figure 3b does not show an increase across development only that at age 16 the effect of the NonCog PGS is greater than it is for ages 7, 9, and 12. Upon consulting Supplementary Table 16 row 238 to row 249 I see that for English and mathematics there is no increase in the prediction of the NonCog PGS until the age of 16.

Have the authors considered that the NonCog PGS is simply capturing more variance associated with cognitive ability rather than noncognitive skills? We read that the phenotypic correlation between the latent measure of cognitive and non-cognitive skills increases with age (Supplementary Table 4), and that the genetic correlation between childhood IQ (range 6 – 18 years of age) and non-cognitive skills is $r_g = 0.45$ (Supplementary Table 15). Furthermore, Supplementary Table 16 shows that the NonCog PGS has the greatest point estimate for prediction at age 16 when compared to younger ages so it would seem plausible.

Response: We thank the reviewer for this comment. We have thought of different ways in which this issue could be addressed. First, we would like to refer Reviewer 3 to our response to Reviewer 2's point 6 above where we present the genetic correlations between noncognitive factors and general cognitive ability over development. These correlations were mostly stable developmentally, and particularly between ages 12 and 16, where we see the sharpest increase in the NonCog PGS prediction of academic achievement. These correlations suggest that noncognitive factors at age 16 do not capture more genetic variance that is shared with general cognitive ability if compared to younger ages.

We also thought of an alternative and potentially more direct way of testing this issue. We re-ran our Cog and NonCog PGS predictions of academic achievement also including phenotypic general cognitive ability (g) in our multivariate models. This very stringent test allowed us to examine whether the noncognitive PGS could still predict academic achievement at age 16 even after accounting for all the variance (not only genetic) shared with general cognitive ability. The results of our PGS analyses are reported in Supplementary Note 3b and in the table below. Although the effect of the prediction was attenuated, the NonCog PGS remained a significant predictor of variation in academic achievement at age 16.

Table S3.1: Noncognitive polygenic score prediction of academic achievement over development accounting for phenotypic general cognitive ability (g).

Measure of achievement	Measure of g	β	Robust standard error	p value
Achievement age 7	g age 7	0.06472661	0.01377481	2.62E-06
Achievement age 9	g age 9	0.08331642	0.01697248	9.16E-07
Achievement age 12	g age 12	0.07146178	0.01742392	4.11E-05
Achievement age 16	g age 16	0.1629107	0.01516482	6.42E-27

Note: All multivariate regression included the following predictors (in addition to the NonCog PGS and g): Cog PGS, the first 10 principal components of ancestry, genotyping chip. All variables were residualized for age and sex and the residuals were standardized before regression analyses.

We now include these additional analyses in the Supplementary Material, and we discuss them in the revised manuscript (see excerpt below).

"In Supplementary Note 3a, we show that this increase in prediction was significant overtime for the NonCog PGS, but not for the Cog PGS. Furthermore, we show that this increase was not explained by the NonCog PGS capturing more cognitive variance later in adolescence (Supplementary Note 3b). [Page 10 of the revised manuscript]

Remarks to the Author:

This is a very through answer and completely addresses my concerns.

2. Similar claims are made with the within vs. between family comparisons. In the between family analysis there doesn't seem to be a consistent change over time only that the age 16 estimate is far higher than at earlier ages. In the within family analysis it is at best ambiguous as to whether there is any increase at from age 7 to age 16. Importantly, Supplementary Table 16 shows that the NonCog PGS predicts the most variance in family SES, more so than the achievement variable described. Taking this information together it would seem that NonCog PGS best captures family SES (Supplementary Table 16 rows 305 and 306 show that the highest r^2 for the NonCog PGS are with family SES) and family SES is the variable responsible for the increase in the predictive power of the NonCog PGS at age 16. When differences the effects of family SES are removed as is the case for the within family analysis in Figure 3C. this increase in the predictive power of the NonCog PGS is also removed. If the NonCog PGS is simply providing a measure of the importance of family SES to achieving after the effects of the NonCog are controlled for this would change the conclusions of the paper.

Response: Thank you for raising this point. To address the Reviewer's concern, we have expanded on our response to Reviewer 1's last comment and extended our analyses to explicitly test whether there is an increase in the association between Cog and NonCog PGS with academic achievement over time after adjusting for SES. These analyses indicated that even after adjusting for family SES, the developmental increase in the NonCog PGS prediction was significant, while that was not the case for the Cog PGS. These analyses are reported in Supplementary Note 3c and in the revised manuscript (see excerpt below).

"In Supplementary Note 3a, we show that this increase in prediction was significant overtime for the NonCog PGS, but not for the Cog PGS. Furthermore, we show that this increase was not explained by the NonCog PGS capturing more cognitive variance later in adolescence (Supplementary note 3b), nor by socio-economic status (Supplementary Note 3c)." [Page 9 of the revised manuscript]

Remarks to the Author:

This answer resolves the issue.

Final Decision Letter: